# SCALABLE GENERATIVE MODELS FOR GRAPHS WITH GRAPH ATTENTION MECHANISM

## ABSTRACT

Graphs are ubiquitous real-world data structures, and generative models that approximate distributions over graphs and derive new samples from them have significant importance. Among the known challenges in graph generation tasks, *scalability* handling of large graphs and datasets is one of the most important for practical applications. Recently, an increasing number of graph generative models have been proposed and have demonstrated impressive results. However, scalability is still an unresolved problem due to the complex generation process or difficulty in training parallelization. In this paper, we first define scalability from three different perspectives: number of nodes, data, and node/edge labels. Then, we propose GRAM, a generative model for graphs that is scalable in all three contexts, especially in training. We aim to achieve scalability by employing a novel graph attention mechanism, formulating the likelihood of graphs in a simple and general manner. Also, we apply two techniques to reduce computational complexity. Furthermore, we construct a unified and non-domain-specific evaluation metric in node/edge-labeled graph generation tasks by combining a graph kernel and Maximum Mean Discrepancy. Our experiments on synthetic and real-world graphs demonstrated the scalability of our models and their superior performance compared with baseline methods.

## 1 INTRODUCTION

Graphs are ubiquitous and fundamental data structures in the real world, appearing in various fields, such as social science, chemistry, and biology. Moreover, in applications such as drug discovery and network simulation, graph generative models that can approximate distributions over graphs on a specific domain and derive new samples from them are very important. Compared with generation tasks for images or natural language, graph generation tasks are significantly more difficult; this is due to the necessity of modeling complex local/global dependencies among nodes and edges as well as the intractable properties of graphs themselves, such as discreteness, variable number of nodes and edges, and uncertainty of node ordering.

Among the several challenges involved in graph generation tasks described above, *scalability* is one of the most important for applications in a wide range of real-world domains. In this paper, we define scalability from three perspectives: *graph scalability*, *data scalability*, and *label scalability*. Graph scalability denotes scalability to large graphs with many nodes within the limit of practical time/space complexity, especially in training. Data scalability denotes scalability to large datasets containing many data. Label scalability denotes scalability to graphs with many node/edge labels. Besides these, it is possible to consider other viewpoints, such as edge number, graph diameter, and the total number of nodes in a dataset. However, because these are closely related to the above three defined perspectives, we consider them as a mutually exclusive and collectively exhaustive division.

In recent years, an increasing number of graph generative models based on machine learning have been proposed. They have demonstrated great performances in several tasks, such as link prediction, molecular property optimization, and network structure optimization (Li et al., 2018; You et al., 2018b; Grover et al., 2018; Luo et al., 2018; You et al., 2018a; Liu et al., 2018; Simonovsky & Komodakis, 2018; Wang et al., 2017; Li et al., 2018). However, to the best of our knowledge, no proposed model is scalable in all three contexts. For example, DeepGMG (Li et al., 2018) can generate only small graphs, and GraphRNN (You et al., 2018b) cannot consider node/edge labels. Besides,

both models have weak compatibility with parallel training, which is key to efficient training on a large dataset.

In this work, we propose the *Graph Generative Model with Graph Attention Mechanism* (GRAM) for generating graphs that is scalable in all three contexts, especially during training. Given a set of graphs, our model approximates their distribution in an unsupervised manner. To achieve graph scalability, we employ an autoregressive sequential generation process that is flexible to variable nodes and edges, and formulate the likelihood of graphs in a simple manner to simplify the generation process. Besides, we apply two techniques to reduce computational cost: breadth-first search (BFS) and zeroing attention weights in edge estimation. Regarding data scalability, we apply a novel graph attention mechanism that simply extends the attention mechanism used in natural language processing (Vaswani et al., 2017) to graphs. This graph attention mechanism does not include sequentially dependent hidden states as the recurrent neural network (RNN) does, which improves the parallelizability of training significantly. Compared with other graph attention mechanisms (Veličković et al., 2018; Abu-El-Haija et al., 2018; Ishiguro et al., 2019), ours is architecturally simple, computationally lightweight, and general, i.e., its applicability is not limited to generation tasks. Finally, for label scalability, we formulate the likelihood of graphs assuming multiple node/edge labels. Also, we use graph convolution/attention layers, whose numbers of parameters do not depend directly on the number of labels.

Moreover, we introduce a unified non-domain-specific evaluation metric for generation tasks of node/edge-labeled graphs. Because such a method does not currently exist, prior works relied on domain-specific metrics or visual inspection, which made unified and objective evaluations difficult. Thus, we construct a unified evaluation metric that dispenses with domain-specific knowledge and considers not only the topology but also node/edge labels by combining a graph kernel and Maximum Mean Discrepancy (MMD) (Gretton et al., 2012). Although a similar statistical test based on MMD via a graph kernel was used for schema matching of protein graphs in Kriegel et al. (2006), to the best of our knowledge, this work is the first to use as an evaluation metric for graph generation tasks.

Our experiments on synthetic and real-world graphs demonstrated that our models can scale up to handle large graphs and datasets that previous methods had difficulty with; our models demonstrated superior results to those of baseline methods.

To summarize, the contributions of this work are as follows:

- We propose GRAM, a graph generative model that is scalable, especially in training.
- We propose a novel graph attention mechanism that is general and architecturally simple as a key portion of the generative model.
- We define scalability in graph generation from three perspectives: number of nodes, data, and labels.
- We construct a unified non-domain-specific evaluation metric for node/edge-labeled graph generation tasks by combining a graph kernel and MMD.

## 2 RELATED WORK

Although there are several traditional graph generative models (Erdös & Rényi, 1959; Albert & Barabási, 2002; Leskovec et al., 2010; Robins et al., 2007; Airoldi et al., 2009), we focus here on recent machine learning-based models, which have outperformed traditional models in various tasks.

In terms of their generation process, existing graph generative models can be classified into at least two types: tensor generation models and sequential generation models. Tensor generation models (Simonovsky & Komodakis, 2018; De Cao & Kipf, 2018; Grover et al., 2018) generate a graph by outputting tensors that correspond to the graph. These models are architecturally simple and easy to optimize for small graphs. However, they face difficulty in generating large graphs owing to the non-unique correspondence between a graph and tensors, limitations in the pre-defined maximum number of nodes, and the increasing number of parameters depending on the maximum graph size. In contrast, sequential generation models, such as DeepGMG (Li et al., 2018), GraphRNN(You et al., 2018b), generate graphs by adding nodes and edges one-by-one, which alleviates the above problems and generates larger graphs. To achieve graph scalability, we employ the latter. However,

to generate a graph with $n$ nodes and $m$ edges, DeepGMG requires at least $\mathcal{O}(mn^2)$ operations because of its complex generation process. GraphRNN reduces this time complexity to $\mathcal{O}(nM)$ with a dataset-specific value $M$ by utilizing the BFS. However, it cannot handle node/edge labels, mainly because it omits the feature extraction process, and relies mainly on the information in the hidden states of the RNN. Moreover, these models include sequentially dependent hidden states, which make training parallelization difficult. In contrast, our model employs a graph attention mechanism without sequentially dependent hidden states. This improves the parallelizability of training significantly. Also, by applying two approximation methods, we reduce the complexity to almost a linear order of $n$ while conducting a rich feature extraction. A comparison of our models' and the baselines' complexities is given in Table 2 in Section A.1.

As another division, there are unsupervised learning approaches and reinforcement learning approaches. Given samples of graphs, unsupervised learning models (Li et al., 2018; You et al., 2018b; Simonovsky & Komodakis, 2018; Grover et al., 2018) approximate the distribution of them in an unsupervised manner. Reinforcement learning models (Luo et al., 2018; You et al., 2018a; De Cao & Kipf, 2018; Liu et al., 2018; Li et al., 2018) learn to generate graphs to maximize a given return, such as QED (Bickerton et al., 2012). Although reinforcement learning approaches have demonstrated promising results on several tasks, such as molecular generation and network architecture optimization, we employ an unsupervised approach; this is because it is considered to be more advantageous when new samples that share certain properties with training samples are required or when the reward functions cannot be designed (e.g., a family of pharmaceutical molecules against a certain disease is known, but the mechanism by which they work is not entirely known).

## 3 PROPOSED METHOD

An outline of this section is as follows: In Section 3.1, we first describe notations and define the considered graph generation task. In Section 3.2, we formulate the likelihood of graphs. In Section 3.3, we present our proposed graph attention mechanism, which is the key portion of our model and is used in feature extraction and edge estimation. In Section 3.4, we present our scalable graph generative model, GRAM. We describe two techniques to reduce the computational cost of edge estimation: BFS in Section 3.5.1 and zeroing attention weights in Section 3.5.2. In Section 3.6, we describe the training and propose an evaluation metric.

### 3.1 NOTATIONS AND PROBLEM DEFINITION

In this work, we define a graph $G = (V, E)$ as a data structure that consists of a node set $V = \{v_1, ..., v_n\}$ and an edge set $E = \{e_{i,j} | i, j \in \{1, ..., n\}\}$. For simplicity, we focus on undirected graphs. We assume nodes and edges are associated with node/edge labels and denote the number of them as $a$ and $b$, respectively. For graph representation, we employ a tensor representation. Specifically, given node ordering $\pi$, we represent a pair of a graph and node ordering $(G, \pi)$ as a pair of tensors $(X^\pi, A^\pi)$. Note that we assume the existence of node ordering in graphs, and $\pi$ is a permutation function over $\{1, ..., n\}$. $X^\pi \in \{0, 1\}^{n \times a}$, whose $i$-th row is a one-hot vector corresponding to the label of $v_{\pi(i)}$, stores information about nodes. $A^\pi \in \{0, 1\}^{n \times n \times b}$, whose $(i, j)$-th element is a one-hot vector corresponding to the label of $e_{\pi(i), \pi(j)}$, stores information about edges. If no edge exists between $v_{\pi(i)}$ and $v_{\pi(j)}$, we replace it with a zero vector. An example is illustrated in Figure 3 in Section A.2. Note that $(G, \pi)$ and $(X^\pi, A^\pi)$ are in unique correspondence. Finally, for simplicity, we do not consider self-looping or multiple edges. A possible extension to self-looping edges would involve adding a step to estimate them. For multiple or directed edges, we can prepare additional labels that correspond to them. In the remainder of this paper, for clear notation, we assume $\pi(i) = i$ and omit the superscript $\pi$.

The target graph generation task is as follows: given samples from a distribution of graphs $p(G)$, approximate the distribution ($\hat{p} \approx p$) such that we can derive new samples from it ($G \sim \hat{p}(G)$).

### 3.2 LIKELIHOOD FORMULATION OF GRAPHS

To approximate distributions over graphs in an autoregressive manner, we formulate the likelihood of a pair of a graph and node ordering $(G, \pi)$ and decompose it into a product of conditional probabilities. Because $(G, \pi)$ uniquely defines $(X, A)$ and vice versa, we have $p(G, \pi) = p(X, A)$, which

is decomposed into a product of conditional probabilities as

$$p(X, A) = p(X_1, A_{1,1}) \prod_{s=2}^{n} p(X_s | X_{<s}, A_{<s,<s}) \prod_{t=1}^{s-1} p(A_{t,s} | A_{<t,s}, X_s, X_{<s}, A_{<s,<s}) \quad (1)$$

where $A_{<s,<s} \in \mathbb{R}^{s-1 \times s-1 \times b}$ represents a partial tensor $A_{i,j}$ $(1 \leq i, j < s)$, and other notations follow this. For simplicity, we omit the slice notation for the last dimension of each tensor. In addition, $(X_{<s}, A_{<s,<s})$ uniquely defines a subgraph of $G$, which we denote as $G_{<s}$. With this formulation, the likelihood of a graph $G$ is defined as marginal, $p(G) = \sum_\pi p(G, \pi) = \sum_\pi p(X, A)$. Also, we represent $(X_{<s}, A_{<s,<s})$ as $G_{<s}$ in the following equations.

During training, given samples from $p(G)$, our model approximates the joint probability $p(X, A)$. More precisely, it approximates $p(X, A)$ by approximating the conditional probabilities $p(X_s | G_{<s})$ and $p(A_{t,s} | A_{<t,s}, X_s, G_{<s})$. As for $p(X_1, A_{1,1})$, we use a sampling distribution from the training data. Although the choice of $\pi$ provides room for discussion, we adopt the BFS used in You et al. (2018b).

On inference, we sequentially sample $X_s$ and $A_{t,s}(s = 2, ..., n, t = 1, ..., s-1)$ from the approximated distribution, and we get $(X, A)$ when the EOS is output. This can be viewed as sampling from the approximated distribution $(G, \pi) \sim \hat{p}(G, \pi)$. In particular, by focusing only on $G$, we can view it as sampling from the marginal $G \sim \hat{p}(G)$.

### 3.3 GRAPH ATTENTION MECHANISM

The aim of employing a graph attention mechanism is to efficiently take in the information of distant nodes by attention. Our graph attention mechanism extends the attention mechanism used in natural language processing (Vaswani et al., 2017) to graphs. However, one of the significant differences between graphs and sentences is that we cannot define absolute coordinates in a graph, which makes it difficult to embed positional information in nodes as in Vaswani et al. (2017). To alleviate this problem, we focus on multi-head attention as in Vaswani et al. (2017), and we use relative positional information between nodes: we introduce bias terms in projection to subspaces, which are functions of the shortest path length between two nodes. The operation of graph attention is illustrated in Figures 5 and 6 in Section A.4.

Following Vaswani et al. (2017), we denote matrices into which $n$ query, key, and value vectors are stacked as $Q = (\boldsymbol{q}_1, ..., \boldsymbol{q}_n)^T (\in \mathbb{R}^{n \times d_Q})$, $K = (\boldsymbol{k}_1, ..., \boldsymbol{k}_n)^T (\in \mathbb{R}^{n \times d_K})$, and $V = (\boldsymbol{v}_1, ..., \boldsymbol{v}_n)^T (\in \mathbb{R}^{n \times d_V})$, respectively. Note that we use $V$ to represent the value matrix, not node set. Also, we assume that the $i$-th row of each matrix corresponds to the feature vector of node $v_i$. Note that the operation of graph attention is permutation invariant. Denoting the dimensions of the output vector and the subspace as $d_O$ and $d_S$, respectively, the graph attention mechanism is defined as

$$\text{GMultiHead}(Q, K, V) = \text{Concat}(\text{head}_1, ..., \text{head}_H)W^O \quad (\in \mathbb{R}^{n \times d_O}) \quad (2)$$

$$\text{where } \text{head}_h = \text{GAttention}(Q, K, V) \quad (\in \mathbb{R}^{n \times d_S}) \quad (h = 1, ..., H)$$

where $H$ represents the number of projections to $d_S$-dimensional subspaces. The operation of $\text{GAttention}(\cdot, \cdot, \cdot)$ is defined as

$$\text{GAttention}(Q, K, V) = (\boldsymbol{o}_1, ..., \boldsymbol{o}_n)^T \quad (\in \mathbb{R}^{n \times d_S}) \quad (3)$$

$$\text{where } \boldsymbol{o}_i = \sum_{j=1}^{n} (\boldsymbol{c}_i)_j (W^V \boldsymbol{v}_j + \boldsymbol{b}^V(v_i, v_j)) \quad (\in \mathbb{R}^{d_S}) \quad (i = 1, ..., n)$$

Each attention weight vector $\boldsymbol{c}_i$ is calculated as

$$\boldsymbol{c}_i = \text{softmax}(\boldsymbol{s}_i) \quad (\in \mathbb{R}^n) \quad (i = 1, ..., n) \quad (4)$$

$$\text{where } (\boldsymbol{s}_i)_j = d_K^{-\frac{1}{2}} (W^Q \boldsymbol{q}_i + \boldsymbol{b}^Q(v_i, v_j))^T (W^K \boldsymbol{k}_j + \boldsymbol{b}^K(v_i, v_j)) \quad (\in \mathbb{R}) \quad (j = 1, ..., n)$$

The parameters to be learned are four weight matrices – $W^Q(\in \mathbb{R}^{d_S \times d_Q})$, $W^K(\in \mathbb{R}^{d_S \times d_K})$, $W^V(\in \mathbb{R}^{d_S \times d_V})$, $W^O(\in \mathbb{R}^{H d_S \times d_O})$ – and three bias terms – $\boldsymbol{b}^Q, \boldsymbol{b}^K, \boldsymbol{b}^V(\in \mathbb{R}^{d_S})$. We used different parameters for each head.

To consider the geometric relation between two nodes, we used functions of the shortest path length between $v_i$ and $v_j$ as $\boldsymbol{b}^Q(v_i, v_j), \boldsymbol{b}^K(v_i, v_j)$, and $\boldsymbol{b}^V(v_i, v_j)$. Furthermore, because path length is discrete, we used different weight parameters for each path length. Examples of other possible approaches include using network flow or functional approximation. Specifically, setting $\boldsymbol{b}^Q = \boldsymbol{b}^K = \boldsymbol{b}^V = \boldsymbol{0}$ yields the original multi-head attention. As in Vaswani et al. (2017), we added a two-layer feedforward neural network (FNN), which is applied after the above operations. We denote these operations, including the FNN, as the graph attention mechanism in the following sections.

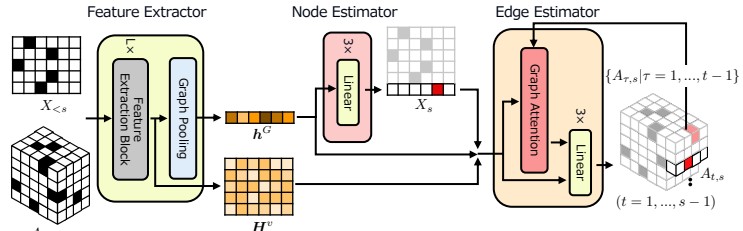

Figure 1: $s$-th step of the generation process.

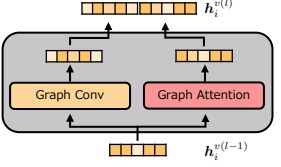

Figure 2: $l$-th feature extraction block.

## 3.4 GRAM: Scalable Generative Models for Graphs

GRAM approximates a distribution over graphs in an autoregressive manner. It utilizes the graph attention mechanism to improve the parallelizability of training. The $s$-th step of the generation process is illustrated in Figure 1; a larger and more detailed version is given in Figure 4 in Section A.3. A detailed analysis of computational complexity is included in Section A.1.

### 3.4.1 Model Overview

In the following sections, we focus on the $s$-th step, i.e., the generation of $v_s$ and $\{e_{1,s}, ..., e_{s-1,s}\}$. The architecture of GRAM consists of three networks: feature extractor, node estimator, and edge estimator. The feature extractor calculates node feature vectors and a graph feature vector by summing them. The node estimator predicts the label of the new node $v_s$. The edge estimator predicts the labels of new edges $\{e_{1,s}, ..., e_{s-1,s}\}$ between previously generated nodes $\{v_1, ..., v_{s-1}\}$ and the new node $v_s$. In other words, the node estimator approximates $p(X_s|G_{<s})$, and the edge estimator approximates $p(A_{t,s}|A_{<t,s}, X_s, G_{<s})$ in Equation 1.

### 3.4.2 Feature Extractor

Given a pair $(X_{<s}, A_{<s,<s})$, the feature extractor calculates $s-1$ node feature vectors $H^v = (\boldsymbol{h}_1^v, ..., \boldsymbol{h}_{s-1}^v)^T$ and a graph feature vector $\boldsymbol{h}^G$ of the corresponding subgraph $G_{<s}$. Here, $\boldsymbol{h}_i^v$ denotes the feature vector of node $v_i$, and we use $\boldsymbol{h}_i^{v(l)}$ to represent the output vector of the $l$-th block. This consists of $L$ feature extraction blocks and a graph pooling layer. In this work, we used $L = 3$.

A feature extraction block is composed of a graph convolution layer and a graph attention layer stacked in parallel, as illustrated in Figure 2. We aim to extract local information by graph convolution layers and global information by graph attention layers. Although there are various types of graph convolutions, we employed the one used in Johnson et al. (2018).[1] Roughly speaking, this convolutes the features of neighboring nodes and edges into each node and edge in a graph. A graph attention layer operates self-attention, where query/key/value vectors are all node feature vectors. To reduce computational cost, we restricted the range of attention to path length $r$. Also, to exploit low-level features, we stacked degree and clustering coefficients, which are non-domain-specific statistics in graphs, into each input node vector. Calculation of these graph statistics and BFS is required only once before training and is relatively fast compared with training time.

The graph pooling layer computes a graph feature vector $\boldsymbol{h}^G$ by summing all node feature vectors in the subgraph. To improve its expressive power in aggregation, we used a gating network as in Li et al. (2018). Specifically, the operation in the graph pooling layer is defined as

$$\boldsymbol{h}^G = \sum_{i=1}^{s-1} \sigma(g_{pool}(\boldsymbol{h}_i^v))\boldsymbol{h}_i^v \tag{5}$$

where $g_{pool}$ is a two-layer FNN, and $\sigma$ is a sigmoid function.

### 3.4.3 Node Estimator

The node estimator predicts the label of new node $v_s$. More specifically, it predicts $X_s$ from $\boldsymbol{h}^G$ as

$$X_s = \text{softmax}(g_{NE}(\boldsymbol{h}^G)) \qquad (\in \mathbb{R}^{a+1}) \tag{6}$$

where $g_{NE}$ is a three-layer FNN, and the output dimension, including the EOS, is $a + 1$. We terminate the generation process when the EOS is output.

---

[1]Because Johnson et al. (2018) assume directed graphs as input, we ignore $g_o$ in Figure 3 of that paper.

### 3.4.4 EDGE ESTIMATOR

The edge estimator predicts the labels of the new edges $\{e_{1,s}, ..., e_{s-1,s}\}$ between previously generated nodes $\{v_1, ..., v_{s-1}\}$ and the new node $v_s$. More precisely, it predicts $A_{t,s}$ ($t = 1, ..., s-1$) from $\boldsymbol{h}_t^v, \boldsymbol{h}^G, \boldsymbol{h}_s^v$ and previously predicted edge labels in this step as

$$A_{t,s} = \text{softmax}(g_{EE}(\boldsymbol{h}_t^v, \boldsymbol{h}^G, \boldsymbol{h}_s^v, \boldsymbol{h}_{<t}^e)) \quad (\in \mathbb{R}^{b+1}) \quad (t = 1, ..., s-1) \quad (7)$$

where $\boldsymbol{h}_s^v$ is the embedded vector of the predicted label of $v_s$, i.e., $X_s$. The vector $\boldsymbol{h}_{<t}^e$ is calculated by a source-target attention in which the query vector is $\text{Concat}(\boldsymbol{h}_t^v, \boldsymbol{h}_s^v)$ and the key/value vectors are $\{\text{Concat}(\boldsymbol{h}_\tau^v, \boldsymbol{h}_s^v, \boldsymbol{h}_{\tau,s}^e) | \tau = 1, ..., t-1\}$, where $\boldsymbol{h}_{\tau,s}^e$ is an embedded vector of the predicted label of $e_{\tau,s}$, i.e., $A_{\tau,s}$. Thereby, we aim to express the dependency on $A_{<t,s}$ in Equation 1. The operation is illustrated in Figure 6 in Section A.4. We use a three-layer FNN as $g_{EE}$, where the output dimension is $b+1$, including the "no edge" label. When the "no edge" label is output, we do not add an edge and we set $A_{t,s} = \boldsymbol{0}$.

### 3.5 REDUCING COMPUTATIONAL COMPLEXITY IN EDGE ESTIMATION

To generate a graph with $n$ nodes, the edge estimation process requires $\mathcal{O}(n)$ operations for one edge, resulting in $\mathcal{O}(n^2)$ operations for one step and $\mathcal{O}(n^3)$ operations in total. This is a significant obstacle to achieving graph scalability. Here, we present two methods to reduce complexity in edge estimation: BFS and zeroing attention weights; these can be combined independently.

#### 3.5.1 COMPLEXITY REDUCTION VIA BREADTH FRIST SEARCH

As in You et al. (2018b), we utilize the BFS node ordering, but our implementation gives a stricter upper bound. More precisely, when estimating new edge $e_{t,s}$ ($t = 1, ..., s-1$), nodes that can be connected to the new node $v_s$ are restricted to "frontier nodes" $V_f = \{v_\tau | \min(\{i | v_i \text{ is connected to } v_{s-1}\}) \leq \tau \leq s-1\}$ under $\pi = \text{BFS}$ . With this property, we need only consider $V_f$ instead of the whole $\{v_1, ..., v_{s-1}\}$ when estimating $e_{t,s}$. This reduces the complexity of edge estimation to $\mathcal{O}(n\beta^2)$, where $\beta = |V_f|$. A proof and example are included in Section A.5. Finally, we denote this variant as GRAM-B.

#### 3.5.2 COMPLEXITY REDUCTION VIA ATTENTION WEIGHTS ZEROING

As another method, we use an approximation inspired by empirical observation. Specifically, our inspection of each attention weight during estimation of new edge $e_{t,s}$ ($t = 1, ..., s-1$) revealed that the nodes predicted to have no connection to $v_s$ (i.e., $V_{\text{noedge}} = \{v_\tau | A_{\tau,s} = \boldsymbol{0}, \tau = 1, ..., t-1\}$) have near-zero weights, while the nodes predicted to have connection (i.e., $V_{\text{haveedge}} = \{v_\tau | A_{\tau,s} \neq \boldsymbol{0}, \tau = 1, ..., t-1\}$) have larger weights. This suggests that among $\{v_1, ..., v_{t-1}\}$, $V_{\text{haveedge}}$ are important to predict the new edge $e_{t,s}$, while $V_{\text{noedge}}$ are not. Hence, we deterministically set the weights of the latter to zero, which means we do not consider them in graph attention. With this approximation, the complexity is reduced to $\mathcal{O}(n^2\alpha)$, where $\alpha = |V_{\text{haveedge}}|$. In addition, because $\alpha \leq \deg(v_s)$ holds, we can assume that $\alpha \ll n$ when $\deg(v_s) \ll n$, which is often the case in many real-world graphs. A more detailed discussion and an example are included in Section A.6. Finally, we denote this variant as GRAM-A and the combination of the two as GRAM-AB, which has complexity $\mathcal{O}(n\alpha\beta)$.

### 3.6 TRAINING AND EVALUATION

The loss function is the negative logarithm of Equation 1, and the training is done in the usual autoregressive manner. In particular, for one graph, forward/backward propagation of $n$ node estimations and at most $n(n-1)/2$ edge estimations can be processed parallelly (i.e., in $\mathcal{O}(1)$ sequential operations) by using ground-truth labels and proper masking.

To evaluate the quality of generation, evaluation metrics based on MMD with graph statistics were used in You et al. (2018b). However, these metrics cannot consider node/edge labels, and it is difficult to determine which one is important or not. Therefore, we construct a unified evaluation metric *GK-MMD* that can consider not only topology but also node/edge labels by combining a graph kernel and MMD.

MMD is a test statistic to determine whether two sets of samples from the distribution $p$ and $q$ are derived from the same distribution (i.e., whether $p = q$). When its function class $\mathcal{F}$ is a unit ball in a reproducing kernel Hilbert space (RKHS) $\mathcal{H}$, we can derive the squared MMD as

$$\text{MMD}^2[\mathcal{F}, p, q] = \mathbb{E}_{x,x'\sim p}[k(x, x')] - 2\mathbb{E}_{x\sim p, y\sim q}[k(x, y)] + \mathbb{E}_{y,y'\sim q}[k(y, y')] \tag{8}$$

where $k(\cdot, \cdot)$ is the associated kernel function (Gretton et al., 2012).

Graph kernels are kernel functions over graphs, and we used the neighborhood subgraph pairwise distance kernel (NSPDK) (Costa & Grave, 2010), which measures the similarity of two graphs by matching pairs of subgraphs with different radii and distances. Because NSPDK is a positive-definite kernel (Costa & Grave, 2010), it follows that it defines a unique RKHS $\mathcal{H}$ (Aronszajn, 1950). This allows us to calculate the squared MMD using Equation 8.

Because NSPDK considers low/high-level topological features and node/edge labels, GK-MMD can be used as a unified evaluation metric for node/edge-labeled graph generation tasks.

## 4 EXPERIMENTS

We performed experiments on four types of synthetic graphs and three types of real-world graphs with various topologies, graph sizes, dataset sizes, and numbers of node/edge labels. We evaluated the quality of generation and the actual computational cost required.

As synthetic graphs, we used four types of random graphs: grid, lobster, community (Fortunato, 2010), and Barabási-Albert (B-A) (Barabási & Albert, 1999). We generated 700 graphs with numbers of nodes in the range $50 \leq |V| \leq 100$. We split them into training/test/validation data at 500:100:100. To make the problem more realistic, we deterministically labeled nodes and edges according to their properties, such as degree or distance from backbone. Detailed configurations, including model-specific parameters and node/edge labeling, are described in Section A.7.

We used three types of real-world graphs: molecular, ego, and protein. For the molecular graphs, we used 250k samples from the ZINC database (Sterling & Irwin, 2015) provided by Kusner et al. (2017). The ego graphs are ego networks from the Citeseer networks (Sen et al., 2008). The protein graphs represent protein structures with amino acids as nodes (Dobson & Doig, 2003). For the ego and protein datasets, we used data provided by You et al. (2018b).

We compared our models with DeepGMG (Li et al., 2018), GraphRNN, and GraphRNN-S (You et al., 2018b) as recent deep learning-based baselines. We modified the GraphRNN and GraphRNN-S models so that they can output multiple node/edge labels. Because we target the unsupervised learning graph generation task and most of the datasets we used are node/edge-labeled, we did not conduct a comparison with reinforcement learning-based methods or traditional models, which are incapable of handling node/edge labels, except for GraphRNN and its variant.

To evaluate the quality of generation, we calculated squared GK-MMD score between generated graphs and test graphs. We also used the squared MMD score of the three types of graph statistics used in You et al. (2018b): distributions of degree, clustering coefficient, and orbit count; however, these metrics do not consider node/edge labels. A lower MMD score indicates a better quality of approximation in terms of the given statistic. For the molecular graphs, we generated 10k samples and calculated the ratios of generated graphs that are valid as molecules (valid [%]), unique samples (unique [%]), and novel samples, i.e., valid, unique, and unseen in the training data (novel [%]). We also reported training time [hour].

The results are listed in Table 1. We did not train DeepGMG on the ego, protein and molecular datasets due to its impractical training time (estimated to take over 90 hours). Instead, for the molecular dataset, we used generated graphs provided by the author for evaluation. Although Deep-GMG achieved a high valid/unique score and low GK-MMD score in the molecular dataset, it failed for larger graphs. This is likely because its feature extraction method is based only on graph convolution, which gives only local information, so it works well only on small graphs, such as those of molecules. Moreover, its training is slow due to its sequentially dependent hidden states.

GraphRNN and its variant achieved low topological MMD scores for simple topology graphs like grid. However, their performances were poor on other graphs with more complex topologies or rules. This is likely because they do not calculate node/edge features, which limited their expressive

Table 1: Results on synthetic graphs (top) and real-world graphs (bottom). The dataset size $N$, maximum graph size $|V|_{\max}$, and the number of node/edge labels $a, b$ are shown as $(N, |V|_{\max}, a, b)$.

| | Grid (500, 100, 3, 2) | | | | | Community (500, 100, 4, 2) | | | | | B-A (500, 100 , 2, 3) | | | | |
|---|---|---|---|---|---|---|---|---|---|---|---|---|---|---|---|
| | deg. | clus. | orbit | GK | time | deg. | clus. | orbit | GK | time | deg. | clus. | orbit | GK | time |
| DeepGMG | 1.315 | 1e-5 | 0.682 | 0.409 | 15.9 | 1.515 | 1.074 | 1.118 | 0.233 | 9.2 | 1.839 | 1.568 | 1.075 | 0.257 | 9.1 |
| GraphRNN | **5e-4** | **0** | **2e-5** | 0.289 | 1.9 | **0.011** | 0.121 | 0.024 | 0.061 | 3.9 | 0.359 | 0.096 | 0.130 | 0.025 | 3.4 |
| GraphRNN-S | 0.040 | 8e-4 | 0.001 | **0.287** | 1.1 | 0.633 | 0.195 | 0.427 | 0.062 | 2.5 | 0.581 | 0.159 | 0.158 | 0.037 | 2.6 |
| GRAM | 0.105 | 0.002 | 0.013 | **0.287** | 0.8 | **0.011** | 0.025 | **0.009** | **0.016** | 1.0 | **0.034** | 0.009 | **0.024** | **0.015** | 1.0 |
| GRAM-A | 0.314 | 0.007 | 0.099 | 0.295 | 0.7 | 0.031 | **0.004** | 0.038 | **0.016** | 0.9 | 0.061 | 0.094 | 0.045 | **0.015** | 1.0 |
| GRAM-B | 0.029 | 0.002 | 0.011 | **0.287** | 0.7 | 0.029 | 0.056 | 0.015 | 0.019 | 0.9 | 0.063 | **0.005** | 0.031 | **0.015** | 1.0 |
| GRAM-AB | 0.357 | 0.015 | 0.126 | 0.309 | 0.7 | 0.164 | 0.080 | 0.080 | 0.020 | 0.9 | 0.044 | 0.015 | 0.031 | 0.020 | 0.9 |

| | Ego (605, 399, 1, 1) | | | | | Protein (734, 500, 89, 1) | | | | | Molecular (250k, 38, 9, 3) | | | | |
|---|---|---|---|---|---|---|---|---|---|---|---|---|---|---|---|
| | deg. | clus. | orbit | GK | time | deg. | clus. | orbit | GK | time | valid | unique | novel | GK | time |
| DeepGMG | - | - | - | - | - | - | - | - | - | - | 84.5 | 99.1 | 83.8 | 0.0201 | - |
| GraphRNN | 0.117 | 0.615 | 0.043 | 0.028 | 16.0 | **0.040** | 1.256 | 0.506 | 0.011 | 22.4 | 11.1 | **100.0** | 11.1 | 0.0284 | 20.5 |
| GraphRNN-S | 0.306 | 0.337 | 0.314 | 0.067 | 11.4 | 0.523 | 1.247 | 0.123 | 0.019 | 13.7 | 26.2 | 99.9 | 26.2 | 0.0551 | 12.4 |
| GRAM | 0.141 | 0.160 | 0.058 | 0.034 | 2.5 | 0.154 | 0.232 | 0.257 | 0.013 | 6.9 | 92.7 | **100.0** | 92.7 | **0.0198** | 9.5 |
| GRAM-A | 0.139 | 0.268 | 0.062 | 0.031 | 2.0 | 0.224 | 0.569 | 0.425 | 0.011 | 4.9 | **95.0** | **100.0** | **95.0** | **0.0198** | 9.4 |
| GRAM-B | **0.108** | **0.083** | **0.033** | **0.017** | 2.0 | 0.154 | 0.247 | **0.040** | 0.010 | 4.6 | 94.3 | **100.0** | 94.2 | 0.0206 | 8.5 |
| GRAM-AB | 0.269 | 0.182 | 0.249 | **0.017** | 2.0 | 0.046 | **0.049** | 0.111 | 0.010 | 4.7 | 94.9 | 99.8 | 94.8 | 0.0206 | 8.4 |

power. In other words, the feature extraction process is inevitable for the generation of complex graphs.

In contrast, our models achieved overall lower MMD scores on most of the datasets, which means that the distance between the approximated distribution and the test set is relatively small in terms of topological and node/edge label features. Also, from the results on the molecular dataset, we can see that the generation is diverse while capturing the strict valence rule via unsupervised learning. This is likely due to the bias term of graph attention in edge estimation: it enables models to control the valence and type of rings. In terms of training time, there is a significant acceleration compared with the baselines. This is mainly owing to the graph attention mechanism, which reduces the number of sequential operations during training to almost constant and enables efficient parallelized training.

Besides, we can observe the impact of the two computational reduction techniques on the actual computational time. They were also more efficient in terms of memory usage, requiring less than half that of the original model. Surprisingly, although there is generally some trade-off between training speed and performance, GRAM-B performed better than the original model on the grid and ego datasets, and GRAM-A performed better on the molecular dataset. This is likely because each approximated model has its own strong topology for which both performance is boosted and computational cost is reduced. On average, the proposed models performed better on large graphs. More detailed discussions are included in Sections A.5 and A.6.

From the above results, we can see the scalability of our models for large graphs and datasets and multiple node/edge labels, which the baseline methods have difficulty in handling. Note that although our approximated models have some trade-off between performance and training speed, they still achieved superior or competitive results over the baselines in most cases. Also, we found an interesting correlation between GK-MMD score and topological MMD scores or visual similarity, which supports the validity of this metric. The detailed setting of the experiments, additional results, visualization, and an ablation study are included in Section A.7.

## 5 Conclusion

In this work, we tackled the problem of scalability as it is one of the most important challenges in graph generation tasks. We first defined scalability from three perspectives, and then we proposed a scalable graph generative model named GRAM along with some variants. Also, we proposed a novel graph attention mechanism as a key portion of the model and constructed GK-MMD as a unified evaluation metric for node/edge-labeled graph generation tasks. In an experiment using synthetic and real-world graphs, we verified the scalability and superior performances of our models.

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

# A APPENDIX

## A.1 COMPLEXITY ANALYSIS

In this section, we compare the computational complexity of the training of our models with that of existing models. Following Vaswani et al. (2017), we consider two factors: total computational complexity and minimum sequential operations with parallelization. Note that we assume all matrix-vector products can be computed in constant time. Table 2 summarizes the results.

To generate a graph with $n$ nodes and $m$ edges, DeepGMG (Li et al., 2018) requires at least $\mathcal{O}(mn^2)$ operations. Because the initialization of the state of the new node depends on the state in the previous step, DeepGMG's minimum required sequential operations is $\mathcal{O}(n^2)$. On the other hand, GraphRNN (You et al., 2018b) requires $\mathcal{O}(nM)$ operations with a dataset-specific constant value $M$, utilizing the properties of the BFS and relying mainly on the hidden states of the RNN. However, its minimum number of sequential operations is $\mathcal{O}(n + M)$ because of the sequential dependency of the hidden states of the RNN; moreover, it cannot generate node/edge-labeled graphs.

Next, we evaluate GRAM and its variants. We denote the average number of nodes/edges within distance $r$ of one node as $N_r/M_r$ and reuse the $\alpha$ and $\beta$ defined in Section 3.5.

In one step, the feature extractor requires $\mathcal{O}(m + nN_r)$ operations, considering a graph convolution layer requires $\mathcal{O}(m)$, and a graph attention layer requires $\mathcal{O}(nN_r)$. However, comparing the node features in $G_{<s}$ with those in $G_{<s+1}(= G_s \cup v_s \cup \{e_{1,s}, ..., e_{s-1,s}\})$, only nodes within distance $rL$ of $v_s$ change their features. Utilizing this property, we conduct feature extraction only for $rL$-neighboring nodes of the previously added node and reuse features for the other nodes, by which the number of feature extraction operations is reduced to $\mathcal{O}(n(M_{rL} + N_{rL}N_r))$.

Also, the node estimator requires $\mathcal{O}(n)$ operations in total. In edge estimation, GRAM requires $\mathcal{O}(n^3)$ operations, GRAM-B requires $\mathcal{O}(n\beta^2)$, GRAM-A requires $\mathcal{O}(n^2\alpha)$, and GRAM-AB requires $\mathcal{O}(n\alpha\beta)$.

Therefore, the total complexity of GRAM is $\mathcal{O}(n^3 + n(M_{rL} + N_{rL}N_r))$, that of GRAM-B is $\mathcal{O}(n\beta^2 + n(M_{rL} + N_{rL}N_r))$, that of GRAM-A is $\mathcal{O}(n^2\alpha + n(M_{rL} + N_{rL}N_r))$, and that of GRAM-AB is $\mathcal{O}(n\alpha\beta + n(M_{rL} + N_{rL}N_r))$. Note that all the estimation operations of the node estimator and the edge estimator can be parallelized in training because our model has no sequentially dependent hidden states like the RNN does.

From the above analysis, we can see that our models require fewer operations than DeepGMG when $N_r \ll n$ holds, which is the often case, while keeping rich feature extraction. In addition, assuming $N_{rL}$ and $M_{rL}$ are nearly constant and sufficiently smaller than $n$ and $m$, respectively, the total complexities of GRAM-B and GRAM-AB can be regarded as almost linear in $n$, which is competitive with GraphRNN, but ours keep the feature extraction. More importantly, all estimation operations can be parallelized, which facilitates training using multiple computing nodes. Therefore, we can expect graph scalability and data scalability of our models. We also expect label scalability because our models are flexible to a variable number of labels by modifying only the dimensions of the input and output layers.

Table 2: Total computational complexity and minimum sequential operations during training.

| Method | Total Complexity | Minimum Sequential Operations |
|--------|------------------|-------------------------------|
| DeepGMG | $\mathcal{O}(mn^2)$ | $\mathcal{O}(n^2)$ |
| GraphRNN | $\mathcal{O}(nM)$ | $\mathcal{O}(n + M)$ |
| GRAM | $\mathcal{O}(n^3 + n(M_{rL} + N_{rL}N_r))$ | $\mathcal{O}(1)$ |
| GRAM-B | $\mathcal{O}(n\beta^2 + n(M_{rL} + N_{rL}N_r))$ | $\mathcal{O}(1)$ |
| GRAM-A | $\mathcal{O}(n^2\alpha + n(M_{rL} + N_{rL}N_r))$ | $\mathcal{O}(1)$ |
| GRAM-AB | $\mathcal{O}(n\alpha\beta + n(M_{rL} + N_{rL}N_r))$ | $\mathcal{O}(1)$ |

## A.2 EXAMPLE OF TENSOR REPRESENTATION

An example of tensor representation is illustrated in Figure 3.

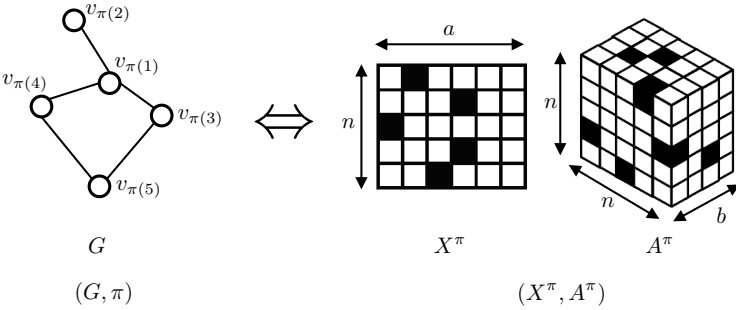

Figure 3: Example of tensor representation. $n, a$, and $b$ denote the numbers of nodes, node labels, and edge labels, respectively. $X^{\pi} \in \{0,1\}^{n \times a}$, whose $i$-th row is a one-hot vector corresponding to the label of $v_{\pi(i)}$, stores information about nodes. $A^{\pi} \in \{0,1\}^{n \times n \times b}$, whose $(i,j)$-th element is a one-hot vector corresponding to the label of $e_{\pi(i),\pi(j)}$, stores information about edges. If no edge exists between $v_{\pi(i)}$ and $v_{\pi(j)}$, we replace it with a zero vector. Note that $(G,\pi)$ and $(X^{\pi}, A^{\pi})$ correspond uniquely.

## A.3 DETAILED MODEL PIPELINE

A detailed pipeline of GRAM is illustrated in Figure 4.

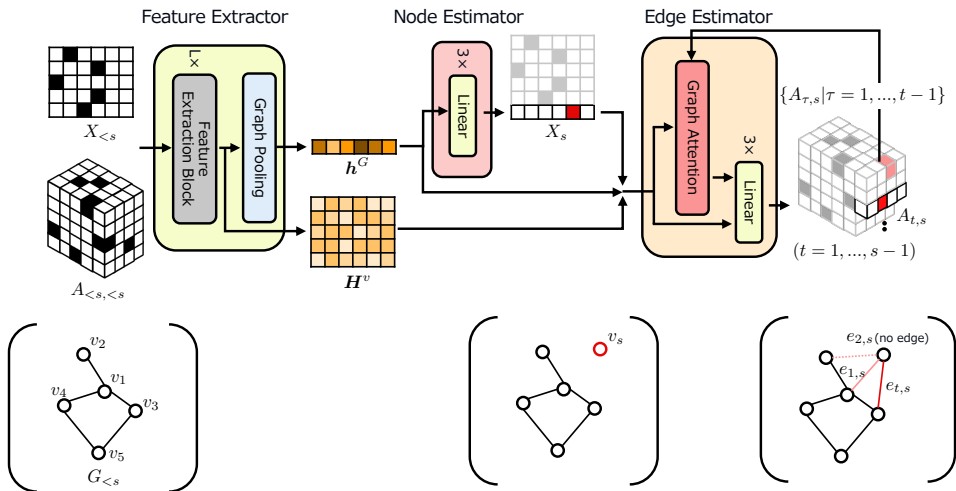

Figure 4: $s$-th step of the generation process ($s = 6, t = 3$). In its $s$-th step, given a subgraph $G_{<s}$, our model estimates the new node $v_s$ and new edges $\{e_{1,s}, ..., e_{s-1,s}\}$ to add next. Specifically, it takes a tensor pair $(X_{<s}, A_{<s,<s})$ as input and predicts the label of the new node $X_s$ and the labels of the new edges $A_{t,s}$ ($t = 1, ..., s - 1$). In detail, first, the feature extractor calculates $s - 1$ node feature vectors $H^v = (\boldsymbol{h}_1^v, ..., \boldsymbol{h}_{s-1}^v)^T$ and a graph feature vector $\boldsymbol{h}^G$ from a tensor pair $(X_{<s}, A_{<s,<s})$. Next, the node estimator predicts the label of the new node $X_s$ from the graph feature vector. When the EOS is output, the generation is terminated. Finally, the edge estimator predicts the labels of the new edges $A_{t,s}$ ($t = 1, ..., s - 1$) from the node feature vectors, the graph feature vector, the embedded vector of the predicted node label, and the embedded vectors of previously predicted edge labels in this step.

## A.4    ILLUSTRATION OF GRAPH ATTENTION

The graph attention mechanism in feature extraction (self attention) is illustrated in Figure 5 and that in edge estimation (source target attention) is illustrated in Figure 6.

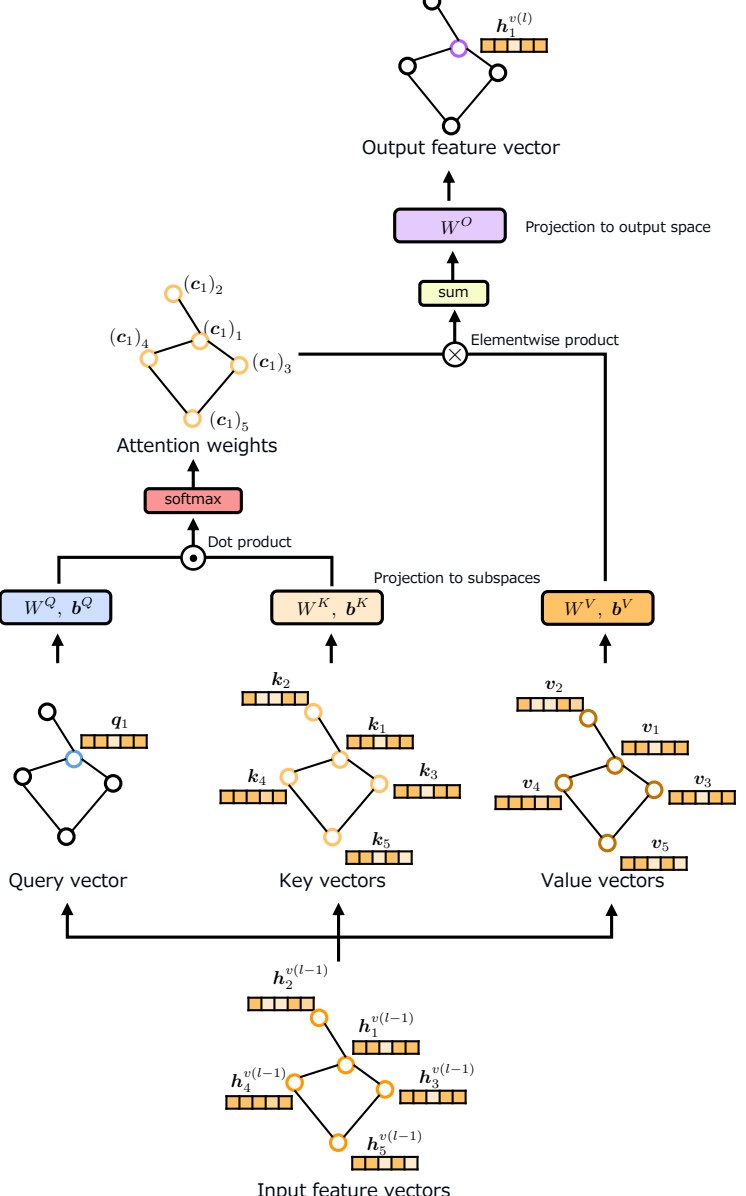

Figure 5: Operation of graph attention in the $l$-th feature extraction block, where query, key, and value vectors are all node feature vectors (i.e., $Q = K = V = H^v = (\boldsymbol{h}_1^{v(l-1)}, ..., \boldsymbol{h}_{s-1}^{v(l-1)})^T)$ (self attention). For simplicity, we focused on one query/output vector and one subspace.

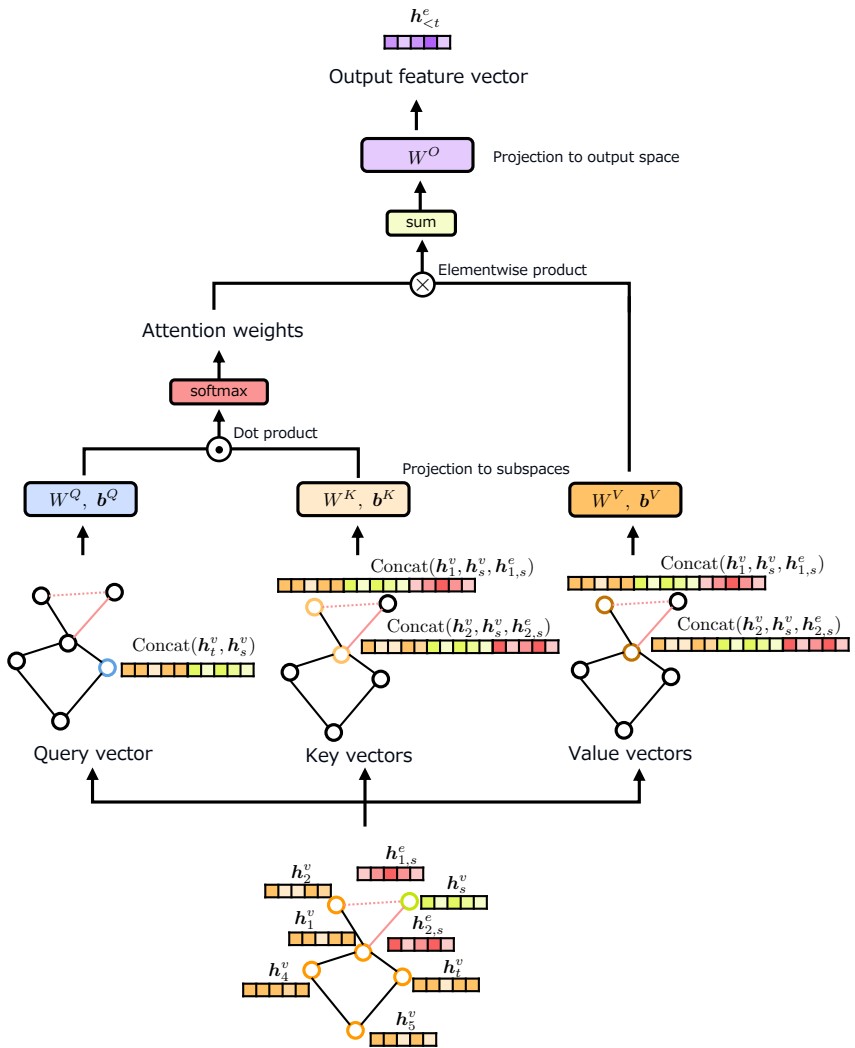

Figure 6: Operation of graph attention in the estimation of the new edge $e_{t,s}$ ($s = 6, t = 3$). The query vector is $\mathrm{Concat}(\boldsymbol{h}_t^v, \boldsymbol{h}_s^v)$, and the value and key vectors are $\{\mathrm{Concat}(\boldsymbol{h}_\tau^v, \boldsymbol{h}_s^v, \boldsymbol{h}_{\tau,s}^e)|\tau = 1,...,t-1\}$, where $\boldsymbol{h}_t^v$ is the feature vector of the node $v_t$, $\boldsymbol{h}_s^v$ is the embedded vector of the predicted label of the new node $v_s$, $X_s$, and $\boldsymbol{h}_{\tau,s}^e$ is the embedded vector of the predicted label of the new edge $e_{\tau,s}$, $A_{\tau,s}$. In other words, $Q = \mathrm{Concat}(\boldsymbol{h}_t^v, \boldsymbol{h}_s^v)^T$, $K = V = (\mathrm{Concat}(\boldsymbol{h}_1^v, \boldsymbol{h}_s^v, \boldsymbol{h}_{1,s}^e), ..., \mathrm{Concat}(\boldsymbol{h}_{t-1}^v, \boldsymbol{h}_s^v, \boldsymbol{h}_{t-1,s}^e))^T$ (source target attention).

### A.5 GRAM-B: UTILIZING BREADTH-FIRST SEARCH NODE ORDERING

In this section, a brief and intuitive proof for the proposition in Section 3.5.1 is given along with additional discussion on how much the computational complexity is actually reduced by this approximation and its strong topology.

#### A.5.1 PROOF

Consider the BFS tree in the left of Figure 7. Suppose there is at least one connection between the new node $v_s$ and the non-frontier nodes $V \setminus V_f$, where $V_f = \{v_\tau | \min(\{i | v_i \text{ is connected to } v_{s-1}\}) \leq \tau \leq s - 1\}$ (right of Figure 7). Then, under the assumption of BFS node ordering, $v_s$ must be visited/generated before $v_{s-1}$, which is a contradiction. Therefore, the nodes that can have connections between the new node $v_s$ are limited to the frontier nodes $V_f$ (middle of Figure 7).

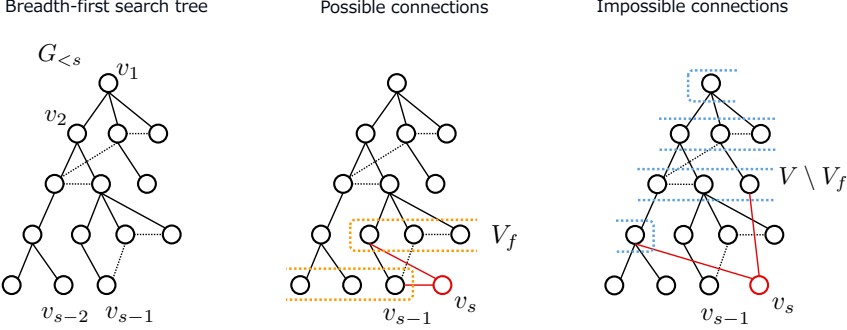

Figure 7: BFS tree example of $G_{<s}$ (left), possible connections to the new node $v_s$ (middle), and impossible connections to the new node $v_s$ (right).

#### A.5.2 DISCUSSION

The average values of $\beta = |V_f|$ for each dataset are listed in Table 3. In both the synthetic and real-world graphs, we can see that $\beta$ is relatively small compared to $n$ (approximately $1/10 \sim 1/3$). Therefore, considering the results in Table 2, the average computational cost of GRAM-B is estimated to be approximately $1/100 \sim 1/9$ that of GRAM, which is a significant reduction.

In the experiment, GRAM-B performed better than its original model on the grid and ego datasets. This is likely because these graphs have simple topologies and their BFS trees also have typical simple patterns. Therefore, restricting the range of edge estimation to the frontier nodes $V_f$ made the problem easier for the model. Thus, we expect GRAM-B's performance to be boosted when the topologies of the graphs and their BFS trees have simple patterns.

Table 3: Average values of $\alpha$ and $\beta$ and the number of nodes $n$ per graph for each dataset.

|  | $\alpha$ | $\beta$ | $n$ |
|---|---|---|---|
| Grid | 1.8 | 9.0 | 72.2 |
| Lobster | 1.0 | 4.2 | 76.0 |
| Community | 2.7 | 23.5 | 74.2 |
| B-A | 3.8 | 31.9 | 72.7 |
| Molecular | 1.1 | 3.6 | 23.2 |
| Ego | 2.1 | 45.8 | 141.1 |
| Protein | 2.5 | 25.7 | 260.2 |

## A.6 GRAM-A: ZEROING ATTENTION WEIGHTS

Here, we provide an additional discussion on the actual computational complexity reduction by this approximation and its strong graph types. Figure 9 illustrates the attention weight zeroing in edge estimation.

Figure 8 shows the two distributions of attention weights in one step of edge estimation (i.e., in the calculation of $h^e_{<t}$) on the grid dataset. The blue represents the distribution of weights for nodes that are predicted to have no connection to the new node $v_s$ (i.e., $\{v_\tau | A^\pi_{\tau,s} = \mathbf{0}, \tau = 1, ..., t-1\}$), and the orange represents the distribution of weights for those predicted to have connection to the new node (i.e., $\{v_\tau | A^\pi_{\tau,s} \neq \mathbf{0}, \tau = 1, ..., t-1\}$). From this figure, we can see that the former distribution is sharper around zero compared with the latter, most of whose weights take near-zero values. Similar results were observed for the other datasets.

The average values of $\alpha$ for each dataset are listed in Talbe 3. In both the synthetic and real-world graphs, we can see that $\alpha$ is less than 4 and approximately $1/100 \sim 1/20$ of $n$. Thus, considering the results in Table 2, the computational cost of GRAM-A is estimated to be approximately $1/100 \sim 1/20$ that of GRAM on average, which is a dramatic reduction.

In the experiment, GRAM-A performed better than GRAM on the molecular dataset. This is likely because these graphs have strict rules or distinct patterns in incident edges to a node, such as the valence rule, and focusing only on the actual generated edges is both sufficient and makes the problem simpler. Thus, we expect that GRAM-A will demonstrate better performance when the graphs follow a certain strict rule on incident edges of their nodes.

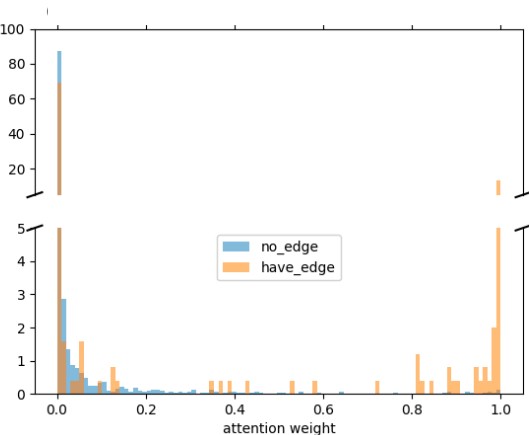

Figure 8: Distributions of attention weights in edge estimation on the grid dataset. The blue represents the distribution of nodes predicted to have no connection to the new node, and the orange represents that of nodes predicted to have connection. More than 90 % of attention weights are less than 0.05 for the former distribution, whereas approximately 70 % of attention weights are less than 0.05 and more than 15 % are larger than 0.95 for the latter. Note that the scale of the y-axis is different between the top and halves.

## A.7 EXPERIMENTAL SETTING DETAILS

### A.7.1 DATASET CONFIGURATION

Here, the detailed configurations of the synthetic graphs are described. Samples from each synthetic graph dataset are illustrated in Figures 10, 11, 12, and 13 (color is best). Different color corresponds to different node/edge labels on each dataset.

**Grid** We used 2D grid graphs with the number of nodes in the range $50 \leq |V| \leq 100$. Each node was labeled according to its degree, yielding 3 types of node labels: "corner", "edge", and

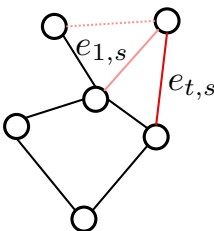

Figure 9: Illustration of attention weight zeroing. In the estimation of $e_{t,s}$, we ignore nodes that are predicted to have no connection with $v_s$ ($v_2$ or $e_{2,s}$ in this case) and set their attention weights to zero deterministically.

"inside". Each edge was labeled according to its geometric direction, yielding 2 types of edge labels: "horizontal" and "vertical".

**Lobster** We used lobster graphs with the number of nodes in the range $50 \le |V| \le 100$. We set probability of adding an edge to the backbone $p_1 = 0.7$, and probability of adding an edge one level beyond the backbone $p_2 = 0.3$. Each node was labeled according to its distance from backbone, yielding 3 types of node labels: "leaf", "branch", and "backbone". Each edge was labeled according to its incident nodes, yielding 2 types of edge labels: "leaf-branch" and "branch-backbone".

**Community** We used 4-community graphs (Fortunato, 2010) with the number of nodes in the range $50 \le |V| \le 100$. We set number of communities $l = 4$, number of nodes in one community $k = |V|/4$, probability of intra-connection $p_{in} = 0.23$, and probability of inter-connection $p_{out} = 0.023$. Each node was labeled according to the community it belongs to, yielding 4 types of node labels: "community1", "community2", "community3", and "community4". Each edge was labeled according to its incident nodes, yielding 2 types of edge labels: "intra-community connection" and "inter-community connection".

**B-A** We used Barabási-Albert (B-A) graphs (Barabási & Albert, 1999) with the number of nodes in the range $50 \le |V| \le 100$. We set number of edges to connect a new node and existing nodes as $m = 4$. Each node was labeled according to whether its degree falls in the top 50% or bottom 50%, yielding 2 types of node labels: "hub" and "exterior". Each edge was labeled according to its incident nodes, yielding 3 types of edge labels: "hub-hub", "hub-exterior", and "exterior-exterior".

### A.7.2 TRAINING CONFIGURATION

Here, we give details of the training configurations. We essentially used the default settings for the baseline methods.

For all datasets, DeepGMG was trained using a single GPU for 10 epochs with batch size 1 because its generation process is so complex that batch implementation is difficult.

GraphRNN and GraphRNN-S were trained using a single GPU for 96000 iterations with batch size 32. For the molecular dataset, considering its relatively large dataset size, we increased the number of iterations to 1536000.

For the convenience of implementation, we started the generation process from seed subgraphs with the number of nodes $N_{\min}$. For the synthetic graph datasets, we trained our models using a single GPU for 100 epochs with batch size 32, and we used $r = 2$ and $N_{\min} = 10$. For the molecular dataset, we trained our models using 2 GPUs for 20 epochs with batch size 256, and we used $r = 7$ and $N_{\min} = 5$. For the ego dataset, we trained our models using 2 GPUs for 25 epochs, and we used $r = 2$ and $N_{\min} = 12$. We used batch size 4 for GRAM and batch size 8 for GRAM-A, GRAM-B, and GRAM-AB. For the protein dataset, we trained our models using 2 GPUs for 25 epochs, and we used $r = 2$ and $N_{\min} = 12$. We used batch size 4 for GRAM and batch size 32 for GRAM-A, GRAM-B, and GRAM-AB. Due to the code dependencies, Tesla P100 was used for DeepGMG, GraphRNN, and GraphRNN-S, and Tesla V100 was used for GRAM and its variants. Empirically, the choice between the two gave no significant difference in computation time for our cases.

In evaluation, we reported an average of 3 runs for each score. To calculate GK-MMD score for the molecular dataset, we used 100 samples from the generated graphs and 100 samples from the test set for fast evaluation. We reported an average of 10 runs.

### A.7.3 ADDITIONAL EXPERIMENT RESULT

The results on the lobster dataset are listed in Table 4. As on other datasets, GRAM and its variants demonstrated competitive results in terms of topological MMDs, and better performance in terms of GK-MMD.

Table 4: Results on lobster dataset. Dataset size $N$, maximum graph size $|V|_{max}$, and the number of node/edge labels $a, b$ are shown as $(N, |V|_{max}, a, b)$.

|  | Lobster (500, 100, 3, 4) | | | | |
| --- | --- | --- | --- | --- | --- |
|  | deg. | clus. | orbit | GK | time |
| DeepGMG | 0.958 | 8e-5 | 0.035 | 0.527 | 13.5 |
| GraphRNN | **1e-4** | **0** | **0** | 0.203 | 2.2 |
| GraphRNN-S | 0.020 | 0.023 | 4e-4 | 0.221 | 1.4 |
| GRAM | 0.001 | 5e-4 | 6e-5 | 0.049 | 0.9 |
| GRAM-A | 0.001 | 0.001 | 8e-4 | **0.043** | 0.8 |
| GRAM-B | 0.001 | 8e-5 | 2e-4 | 0.048 | 0.7 |
| GRAM-AB | 0.001 | 2e-5 | 7e-5 | 0.047 | 0.7 |

In addition, considering that the molecular dataset is a subset of the druglike category in ZINC database (Sterling & Irwin, 2015), we reported the average QED scores (Bickerton et al., 2012) of the training samples and the generated samples. QED score measures the drug likeliness of the molecular.

Table 5 summarizes the results. We can see the generated molecular graphs of our models have higher QED compared with those of baselines, and near to that of the training set. Note that our models do not rely on reinforcement learning methods and utilize neither the information of QED nor valence rule; they learned only through unsupervised learning.

Table 5: Average QED score.

|  | QED |
| --- | --- |
| Train | 0.732 |
| DeepGMG | 0.644 |
| GraphRNN | 0.563 |
| GraphRNN-S | 0.496 |
| GRAM | **0.722** |
| GRAM-A | **0.722** |
| GRAM-B | 0.691 |
| GRAM-AB | 0.694 |

### A.7.4 ABLATION STUDY

To examine the effectiveness of the bias terms in the graph attention mechanism, we conducted an ablation study. Specifically, we evaluated two variants: GRAM without bias terms of graph attention in edge estimation, and GRAM without bias terms of graph attention in feature extraction.

The results are listed in Table 6. On all the synthetic graph datasets, removing bias terms in edge estimation degraded its performance. From this, we can see the importance of the bias terms in edge estimation for the better quality of generation. Also, removing bias terms in feature extraction degraded the performance on the B-A dataset. This is likely because its expressive power is limited.

On the other hand, the drop in the performance by removing bias terms in feature extraction is minor on the grid, lobster and community dataset. Considering the properties of these random graphs, this is likely because only local features were enough for these cases.

Table 6: Results of ablation study.

|  | Grid | | | | Community | | | |
| --- | --- | --- | --- | --- | --- | --- | --- | --- |
|  | deg. | clus. | orbit | GK | deg. | clus. | orbit | GK |
| GRAM | **0.105** | 0.002 | 0.013 | **0.287** | 0.011 | 0.025 | **0.009** | **0.016** |
| w/o bias in feature extraction | 0.111 | **0.001** | **0.009** | **0.287** | **0.006** | **0.007** | 0.016 | **0.016** |
| w/o bias in edge estimation | 0.247 | 0.272 | 0.190 | 0.306 | 0.066 | 0.037 | 0.038 | 0.017 |

|  | B-A | | | | Lobster | | | |
| --- | --- | --- | --- | --- | --- | --- | --- | --- |
|  | deg. | clus. | orbit | GK | deg. | clus. | orbit | GK |
| GRAM | **0.034** | **0.009** | **0.024** | **0.015** | **0.001** | 5e-4 | **6e-5** | **0.049** |
| w/o bias in feature extraction | **0.034** | 0.020 | 0.033 | 0.016 | 0.003 | **1e-4** | 2e-4 | 0.054 |
| w/o bias in edge estimation | 0.050 | 0.035 | 0.042 | 0.017 | 0.002 | 3e-4 | 8e-5 | 0.054 |

### A.7.5 VISUALIZATION OF GENERATED GRAPHS

Visualizations of generated graphs on each dataset are shown in Figures 10, 11, 12, 13 and 14 (color is best). We used the same visualization method for all graphs. Different color corresponds to different node/edge labels on each dataset.

The graphs generated by GraphRNN are quite similar to those in the training set in terms of their topology. However, they completely failed to capture node/edge label patterns in most cases.

In contrast, the graphs generated by GRAM are similar to training samples in terms of both topology and node/edge label pattens.

### A.8 LIMITATIONS AND FUTURE WORK

While our models demonstrated superior performance on most of the datasets, they have some limitations. For example, on inference, one wrong prediction of node/edge label may collapse all the following generation processes, resulting in a bizarre graph generated. This is a deficit of its rich feature extraction process. A possible solution to this problem is, for example, to employ beam search. By applying beam search on inference, we can get rid of samples with a quite lower likelihood and avoid the above undesirable situations.

Additionally, although we reduced the computational complexity of the models by a significant amount, the feature extraction process still requires a large amount of computation. To this challenge, some methods proposed in the field of image processing and natural language processing would help to alleviate it.

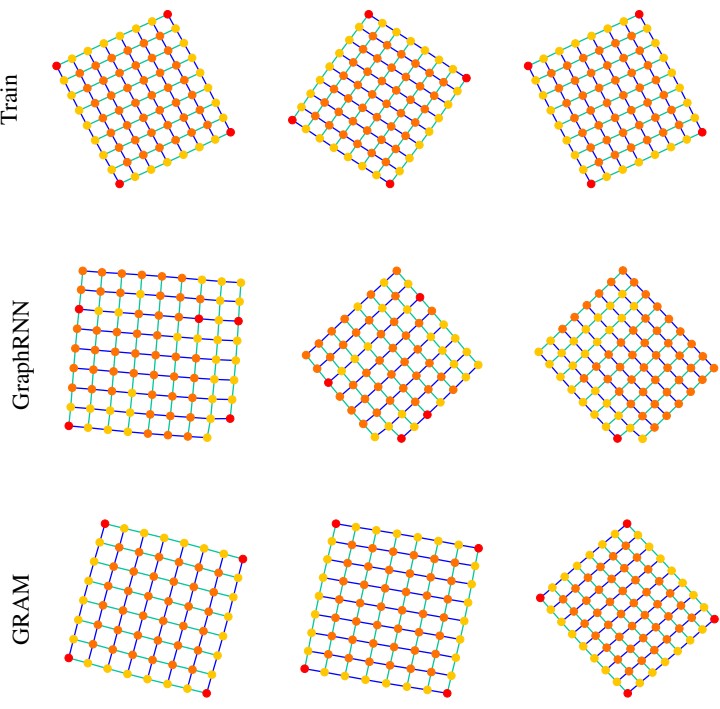

Figure 10: Generated graphs (grid).

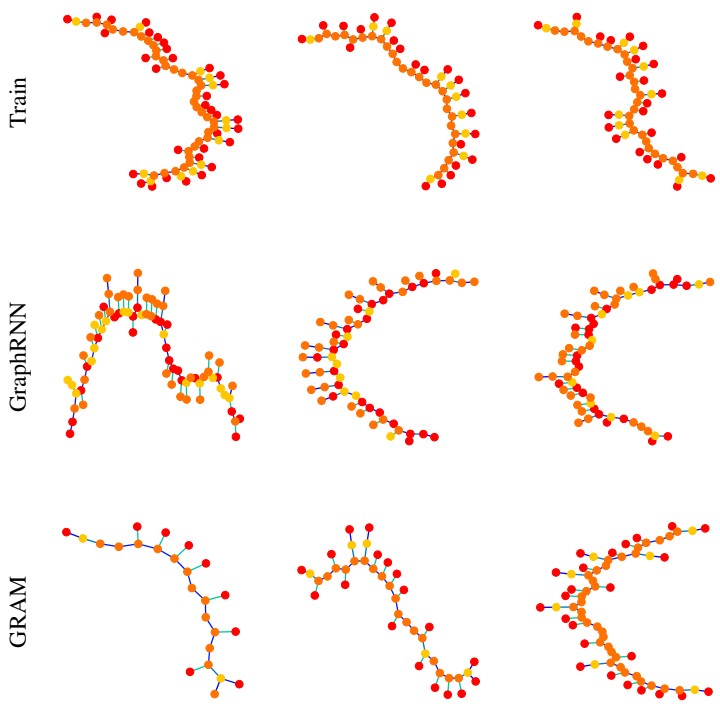

Figure 11: Generated graphs (lobster).

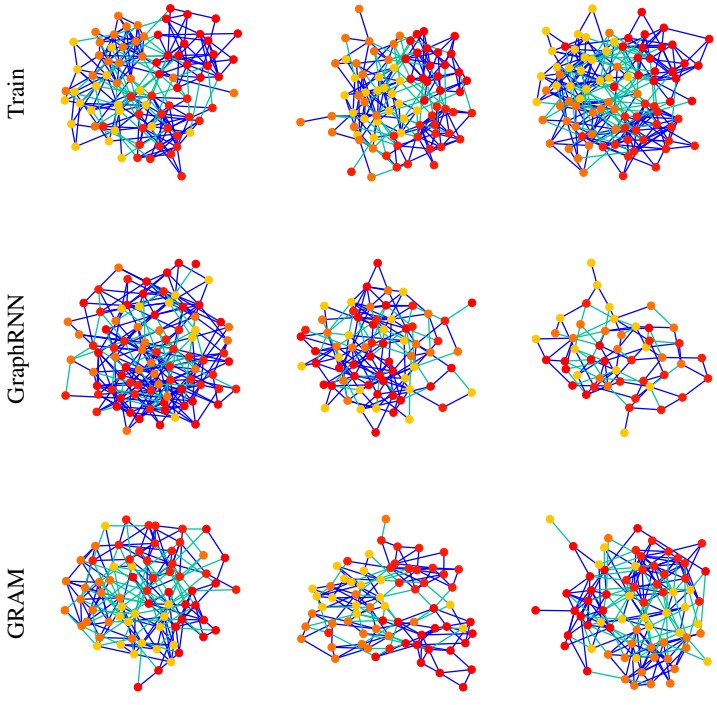

Figure 12: Generated graphs (community).

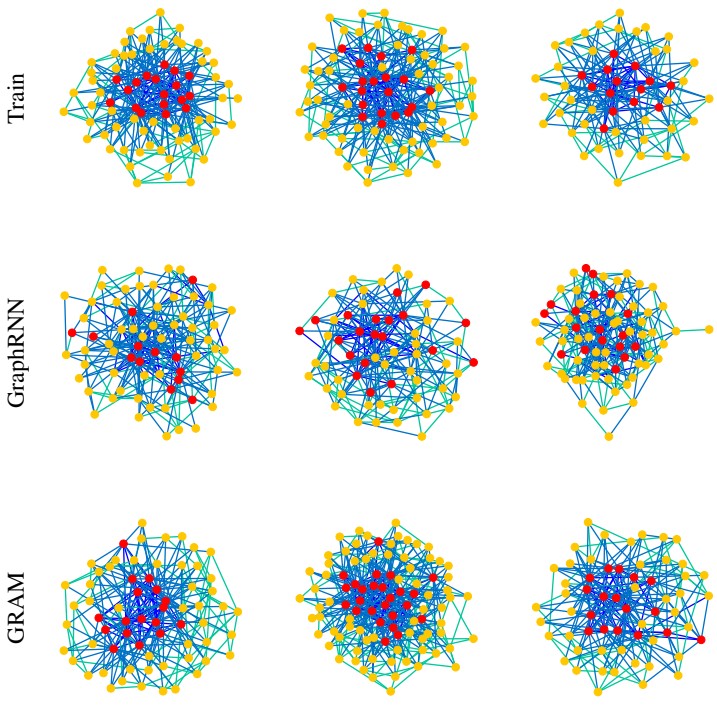

Figure 13: Generated graphs (B-A).

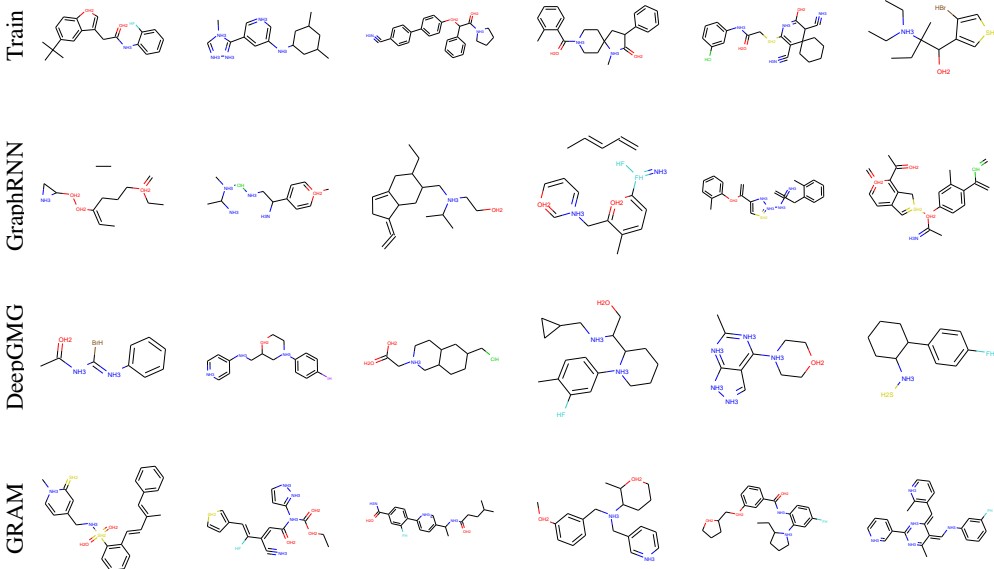

Figure 14: Generated molecular graphs.

