# OpenReview forum: "Scalable Generative Models for Graphs with Graph Attention Mechanism"
_ICLR.cc/2020/Conference — Reject_

### Official Review · AnonReviewer1 · 2019-10-23
**Official Blind Review #1**

**Rating:** 3

**Review:**

# Response to rebuttal

I would like to thank their authors for their rebuttal.

After reading the other reviews, the author response and the revised manuscript, I have decided to keep my score of weak reject for the time being.

In short, while I appreciate the effort the authors put in partly addressing some of the most important comments raised during the review process, I think the paper would greatly benefit from some additional work. In particular:

(1) Given the emphasis on scalability, I still believe the authors should carry out more thorough experiments to characterize the runtime of their approach with respect to different characteristics of the graphs. While the result provided in the response to Reviewer #3 is a first step, I recommend the authors to extend it by (1) varying graph size (in terms of nodes and edges); (2) varying graph type and (3) reporting the speedup with respect to other baselines.

(2) To the best of my knowledge, the ablation experiment in Section A.7.4 does not provide results for the setting in which no graph attention mechanism is used at all, neither for the case where the graph attention mechanism used is identical to GAT (restricted to 1-step neighbourhoods).

(3) While NSPDK might be a reasonable choice, I still am of the opinion that the choice of graph kernel for this purpose is highly arbitrary and, thus, should be investigated further. Given that such a choice is being used to define a performance metric, which moreover is being highlighted as a contribution, the authors should study the robustness of the metric to the choice of graph kernel, as well as its sensitivity to known perturbations.

(4) Finally, I did not see any error bars added to the main results in the paper.

Despite these shortcomings, I would like to reiterate that I believe the proposed approach is promising and, with some additional work, would be a contribution definitely worth publishing. Therefore, I would like to encourage the authors to further revise the manuscript.

# Summary

In this paper, the authors propose an auto-regressive deep generative model for graph-structured data, motivated by the goal of scalability with respect to graph size, graph density and sample size.

In a nutshell, the approach follows closely the ideas in [1, 2], which model graph generation as an auto-regressive process after fixing or sampling an ordering for the nodes. Unlike [1, 2], however, the proposed method makes use of graph convolutions and a graph attention mechanism, closely related to GAT [3], to parametrize the conditional distributions of node/edges given the previously generated graph elements.

The performance of the proposed approach is evaluated in comparison to [1, 2] in several synthetic and real-world datasets, using MMD [4] between generated and held-out test graphs as metric. Unlike [2], which applies MMD on three graph statistics (degree, clustering coefficient and average orbit counts), this manuscript proposes to evaluate MMD using a graph kernel as well [5].

# High-level assessment

The main contribution in this paper is to combine a graph attention mechanism, which can be seen  as a simplification of GAT [3], with deep autoregressive graph models, such as DeepGMG [1] or GraphRNN [2]. In this way, the manuscript has a large conceptual overlap with the method in [6], which can be nevertheless be regarded as concurrent rather than prior work. From a methodological perspective, I believe the contribution is sound and sufficiently novel, although perhaps slightly on the incremental side.

However, the current version of the manuscript has shortcomings regarding (i) lack of clarity in the exposition of the method’s relation to prior work, low-level implementation details and experimental setup and (ii) insufficient experimental results to back up some of the authors’ claims.

Nonetheless, I believe the proposed approach is promising, and encourage the authors to address or clarify these issues during the author discussion phase.

# Major points / suggestions

1. The manuscript presents the proposed approach in a way that does not clearly differentiate between prior work and original contributions.

In particular, I believe that the ideas in Section 3.1 and 3.2 are almost identical to those in [1, 2], the graph attention mechanism in Section 3.3 can be seen as a minor modification of GAT [3], and Section 3.4 also has a strong conceptual overlap with [1, 2].

I would encourage the authors to be more clear with respect to what is novel and what is borrowed from prior work. Moreover, when slightly departing from prior work (e.g. the modifications applied to the graph attention mechanism in Section 3), I would also encourage the authors to focus on explaining what specifically has changed and what is the rationale behind those design choices, rather than explaining the entire mechanism “from scratch”, leaving up to the reader to figure out what is novel.

2. The paper’s clarity could be improved, with some parts presented in an unnecessarily complicated manner (e.g. the graph attention mechanism) and others without sufficient detail (e.g. the edge estimator module, the zero-ing heuristic for attention or the generation of graphs based on “seed graphs”, which is only mentioned in the appendix).

For example, regarding the graph attention mechanism, I would recommend: (i) explaining more clearly what the “feature vector of node $v_{i}$” is exactly in relation to the notation of Section 3.1; (ii) if the query, key and value matrices are identical, as the text seems to imply, I would rewrite the equations directly in terms of $X$ which would simplify the notation significantly; (iii) perhaps most importantly, the bias functions $b^{Q}$, $b^{K}$ and $b^{V}$ should be defined mathematically and discussed in greater detail and (iv) the output FNN should also be described mathematically. Finally, as mentioned above, I would emphasise the differences between the proposed attention mechanism and GAT.

The edge estimator mechanism is described too imprecisely in Section 3.4.4. While Section A.4 definitely helps, I would recommend defining the entire operation mathematically in Section 3.4.4 as well. Likewise, a precise mathematical definition of GRAM-A in Section 3.5.2 would also be helpful.

Finally, as mentioned in this forum by Prof. Ranu prior to this review’s writing, the graph generation procedure described in Section A.7.2 seems unconventional. I would encourage the authors to both clarify what they mean by “for the convenience of implementation” and to investigate whether the experimental conclusions are affected by this departure from prior practices.

3. Key details about the experimental setup, such as the hyperparameter selection protocol for the proposed approach and baselines, as well as the resulting architectures, seems to be missing, making it difficult to assess if the experimental setup is “fair”.

In particular, all methods should be allowed to use a similar number of parameters or, alternatively, have their hyperparameters tuned equally carefully for each dataset separately.

4. Most importantly, I believe the experimental results are insufficient to back up some of the claims made in the introduction.

    4.1. Despite the focus on scalability throughout the motivation, there are no experiments systematically exploring how the runtime at train and test time of the proposed approach and the main baselines scales with respect to sample size, number of nodes per graph and graph density. Moreover, no results are provided for large graphs (e.g. ~5k nodes as in [6]).

    4.2. The graph attention mechanism was claimed to be an original contribution. However, no results are provided to evaluate its advantages with respect to the different GAT variants nor ablation studies to see its usefulness relative to a variant of the proposed approach using only graph convolutions.

    4.3 The idea of using MMD in conjunction with graph kernels as a performance metric is interesting. However, there is no investigation of key aspects such as (i) its relation to other metrics and (ii) the impact that the choice of graph kernel, among the many available, and/or of graph kernel hyperparameters has on the resulting metric (see [7] for a comprehensive review on graph kernels).

   4.4. Finally, the results have been reported without error bars, making it difficult to quantify the statistical significance of the observed performance differences between approaches.

# Minor points / suggestions

1. I strongly believe the authors should adapt the manuscript to mention [6] and related/concurrent work. Ideally, including it as an additional baseline would be even better, but not necessary given the limited rebuttal time. Nevertheless, this point was not taken into consideration when scoring the manuscript, given how recent [6] is.

# References

[1] Li, Yujia, et al. "Learning deep generative models of graphs." *International Conference on Machine Learning.* 2018.
[2] You, Jiaxuan, et al. "Graphrnn: Generating realistic graphs with deep auto-regressive models." *International Conference on Machine Learning.* 2018.
[3] Veličković, Petar, et al. "Graph attention networks." *International Conference on Learning Representations*. 2018.
[4] Gretton, Arthur, et al. "A kernel method for the two-sample-problem." Advances in Neural Information Processing Systems. 2007.
[5] Costa, Fabrizio, and Kurt De Grave. "Fast neighborhood subgraph pairwise distance kernel." Proceedings of the 26th International Conference on Machine Learning. Omnipress; Madison, WI, USA, 2010.
[6] Liao, Renjie, et al. "Efficient Graph Generation with Graph Recurrent Attention Networks." *Advances in Neural Information Processing Systems.* 2019.
[7] Kriege, Nils M., Fredrik D. Johansson, and Christopher Morris. "A Survey on Graph Kernels." *arXiv preprint arXiv:1903.11835* (2019).

**Experience Assessment:**

I have read many papers in this area.

**Review Assessment: Checking Correctness Of Derivations And Theory:**

I carefully checked the derivations and theory.

**Review Assessment: Checking Correctness Of Experiments:**

I carefully checked the experiments.

**Review Assessment: Thoroughness In Paper Reading:**

I read the paper thoroughly.

---

> ### Author Response · Authors · 2019-11-11
> **Author Response for Official Blind Review #1**
>
> To Reviewer #1
>
> Thank you for the detailed review and helpful suggestions.
> Below we answer the clarification points in the review one by one.
> We are preparing and conducting additional experiments/analysis now, and we will report the results as soon as ready.
>
>
> # 1
> Thank you for pointing out. As the reviewer mentions, our work shares some points with previous works. However, we believe it has several different or improved points, which would like to be noted. Below we list the shared points and different points one by one. We will specify them on revising as the reviewer pointed out.
>
> Section 3.1 and 3.2 describe notations and likelihood formulation of graphs. As they are fundamental for graph generation, these parts certainly have common with previous works [1, 2]. On the other hand, these parts differ from previous works in that we formulated the likelihood of graphs in a general manner assuming node/edge labels, as it is necessary for the following model description. This is not explicitly included in those previous works.
>
> Our graph attention shares the basic idea with Compared to GAT [3]: aggregating node features via attention. On the other hand, GAT only considers 1-step neighbors whereas our graph attention can consider $r$-step neighbors ($r > 1$) and embed relative positional information between nodes.  We believe it will be one reliable step for future applicabilities of attention mechanisms in graphs.
>
> The proposed model has in common with previous works [1, 2] in that they generate a graph adding nodes and edges sequentially. The main point our model differs from them is that it employs graph attention mechanism, which reduces the minimum sequential operations during training to O(1) and enables efficient parallelized training. In addition, our model can consider node/edge labels, which GraphRNN [2] cannot handle. We would like it to be noted these points  O(1) minimum sequential operations on training along with the capacity to handle node/edge labels, which are the out of range of prior works.
>
> # 2.
> Thank you for pointing it out. We will modify the parts as mentioned on revising.
>
> # 3.
> Basically, we used the default parameter setting of the prior works. For GraphRNN, we modified the model so that it can output node/edge labels and added parameters accordingly. We will include that on revising.
>
> # 4.2.
> In Section A.7.4, we included an ablation study on the bias terms of our graph attention, which can be considered as a comparison with GAT variants. We will add some description of that on revising.
> # 4.3.
> (i) Since it is usually hard to directory determine a projecting function that corresponds to a specific kernel, it would be difficult to directory relate graph kernel MMD with other MMD scores. However, since the used kernel, NSPDK [4] considers low/high-level structures of graphs, the graph kernel MMD score is expected to measure several graph statistics such as degree, clustering coefficient, and orbit count.
>
> # 4.4.
> Thank you for pointing it out. We will include error bars on revising.
>
> # Minor 1
> Thank you for pointing out. We will include that work in related work.
>
>
> # Reference
> [1] Li, Yujia, et al. "Learning deep generative models of graphs." *International Conference on Machine Learning.* 2018.
> [2] You, Jiaxuan, et al. "Graphrnn: Generating realistic graphs with deep auto-regressive models." *International Conference on Machine Learning.* 2018.
> [3] Veličković, Petar, et al. "Graph attention networks." *International Conference on Learning Representations*. 2018.
> [4] Fabrizio Costa and Kurt De Grave. “Fast neighborhood subgraph pairwise distance kernel.” *International Conference on Machine Learning,* 2010.

---

> > ### Author Response · Authors · 2019-11-15
> > **Author Response 2 for Official Blind Review #1**
> >
> > # 1.
> > Please refer to (5) in the Author Response 2 for Official Blind Review #2.
> >
> > # 4.1.
> > Please refer to Concern and Additional Experiments 2 in the Author Response 2 for Official Blind Review #3.

---

### Official Review · AnonReviewer2 · 2019-10-24
**Official Blind Review #2**

**Rating:** 1

**Review:**

This paper presents a formulation of graph generative models based on graph attention aimed at scalability of these methods.

The paper is generally written well and I like the overall theme of the paper, however, there are a few key issues with this work and I don't think the paper as it stands is ready for publication:

1) The main motivation expressed in the paper is that graph generative models are generally not scalable and they identify three main areas: (a) graph size (i.e. num nodes); (b) data scalability (i.e. num training samples); (c) label scalability (i.e. num of node or edge types). However, the paper doesn't follow on why the proposed method actually addresses these issues. The derivation doesn't talk about scale until we reach section 3.5 and then we find out that actually the proposed model is O(n^3) in reality. Then there are approximations to make it scale. So for me there is a massive disconnect between the main motivation of the paper and the suggested model. Why not study approximation methods for already existing graph generator models?

2) Following on the theme of scale, the only experimental result discussing this is the time column reported for the training time. So that partially addresses the data scalability. Other baselines as well have reasonable training times specially when it comes to large datasets (e.g. ZINC is that the largest dataset studied with 250K samples and GraphRNN is 2x slower and GraphRNN-S only about 20%). What I was looking for was when you really can train on real-world datasets that other methods basically can't be trained. The datasets chosen all are small hence there's not much issue with scale there. The question about the scalability w.r.t. other aspects (i.e. num nodes and num labels) has not been studied or reported.

3) The approximations suggested in section 3.5 also don't seem to have much impact on the training time. These approximations were motivated by the scale while looking at the training times they barely make any difference. However, they make a big difference in performance metrics specially in smaller datasets. So the question that comes to mind is that what is the role of these approximations w.r.t. the quality of the models? Again this question needs further study.

4) Comparing GraphRNN and GraphRNN-S's modifications with the results from the original paper, it seems they are performing much worse (e.g. deg for the original GraphRNN-S is 0.057 while the reported num here is 0.523 for Protein dataset). The same is true for other metrics. Why is that?

5) As pointed out by an observer, it seems that there are nuances to generation of the graph needing seeds of arbitrary size to be provided, explained deep down in the appendix. If this is the case for generation then it should be discussed in the main part of the paper and contrasted with methods that can start from scratch.

6) In the training configuration part of the appendix, A.7.2 it seems there are discrepancies in number of GPUs as well kinds of GPUs used for each method. When reporting training times in the main section, do you normalise against these?

7) It seems that many hyperparameters mentioned in A.7.2 are chosen in an ad-hoc manner without proper model selection and seem to vary across each different versions of GRAM for each different dataset. How sensitive is the model to these hyperparameters? I suspect if the model was insensitive, you could've fixed them for many of these experiments, but seems that is not the case. So without proper model selection routines, the results may not be representative of what the model discussed.

Minor comment: The model suggested has some similarities to DEFactor model from Assouel et al 2019 in terms of formulation of the problem for labelled graphs (nodes as a matrix and adj as a tensor), though the underlying models are very different, that paper as well targets arbitrary size graph generation and efficiency w.r.t. model parameters.

**Experience Assessment:**

I have published one or two papers in this area.

**Review Assessment: Checking Correctness Of Derivations And Theory:**

I assessed the sensibility of the derivations and theory.

**Review Assessment: Checking Correctness Of Experiments:**

I carefully checked the experiments.

**Review Assessment: Thoroughness In Paper Reading:**

I read the paper thoroughly.

---

> ### Author Response · Authors · 2019-11-11
> **Author Response for Official Blind Review #2**
>
> To Reviewer #2
>
> Thank you for the detailed review and helpful suggestions.
> Below we answer the clarification points in the review one by one.
> We are preparing and conducting additional experiments/analysis now, and we will report the results as soon as ready.
>
> # (1)“Why not study approximation methods for already existing graph generator models?”
> Sorry for the insufficient explanation. Our motivation and flow of the paper are 2-fold; (1) reduce the time complexity (i.e., minimum sequential operations) during training to O(1) by graph attention mechanism (Section 3.3 and 3.4) (2) reduce the total complexity from O($n^3$) to O($\alpha n^2$) and O($\beta n$) (Section 3.5).
> We believe that approximating existing models such as DeepGMG and GraphRNN will have limitations in terms of the three scalabilities mentioned in the paper. For example, RNN is their backbone module, which essentially prevents efficient parallelized training. Also, GraphRNN is basically designed for unlabeled graph generation tasks.  To achieve all of the scalabilities, another new model, not an approximation of existing models, is thought to be required inevitably.  We will add the explanation on revising.
>
> # (2)”What I was looking for was when you really can train on real-world datasets that other methods basically can't be trained”
> Our motivation and standpoint are that there are no existing models that can be scalable in the three perspectives at the same time, and can demonstrate promising results. In other words, we would like to mention that although GraphRNN can be trained on the ZINC dataset, its performance is poor. The essential aim of the paper is to demonstrate the scalability of the proposed model theoretically as in Section A.1, not show that impossible was made possible empirically.
> For instance, although oversimplified, the observation that the proposed model is trained on 250k molecular dataset on 2 GPUs for 9 hours would indicate the possibility of more large scale training, such as 10M molecular dataset on 8 GPUs for 90 hours.
>
> # (2)”The question about the scalability w.r.t. other aspects (i.e. num nodes and num labels) has not been studied or reported.”
> In the paper, our criteria to measure the scalability was that (1) whether the model can be trained within practical computational resources, and (2) whether the performance is promising enough, even for large graphs, large datasets, and many node/edge labels. In other words, we intended to use the competitive/high performance on large graphs and graphs with many node/edge labels as the base to state the node/label scalability of the proposed models. We will specify that on revising.
>
> # (3)”The approximations suggested in section 3.5 also don't seem to have much impact on the training time. ”
> Yes, the difference in training time among the approximated models is minor compared with that of baselines. This is thought to because their minimum sequential operations during training are all O(1) thanks to graph attention mechanism, as described in Section 3.6. Instead, these several approximations have impacts on “total” computational cost in edge estimation. Although strict experimental evaluation is difficult, we could increased the batch size 8 times from the base model on the protein dataset, for example. This would indicate their impact on memory efficiency.
> We included some theoretical evaluation of total complexity in Section A.1.
>
> # (3)”So the question that comes to mind is that what is the role of these approximations w.r.t. the quality of the models?”
> We included some discussions on the appropriate selection of approximated models for several types of graphs in Section A.5 and A.6. We will write that in the main paper on revising.
>
> # (4)
> For one thing, this is likely because GraphRNN variants were modified so that they can estimate node/edge labels. In other words, labeled graph generation is thought to be more difficult than unlabeled graph generation, which is one cause for the decrease in the performance. In addition, we removed a post-process before evaluation from the original paper’s code, which appeared to align the sizes of generated graphs. This is for the correct MMD score calculation.
>
> # (5)
> Thank you for pointing out. As the reviewer mentions, we will include that part in the main paper for clarity on revising.
>
> # Minor comment
> Thank you for informing us of that work. We will include it in the related work section on revising.

---

> > ### Author Response · Authors · 2019-11-15
> > **Author Response 2 for Official Blind Review #2**
> >
> > # (5)
> >
> > The proposed model is not designed to generate from scratch as explained in the response (1,2) to the public reviewer. Instead, we conducted experiments aligning seed size to 5.
> > The tables below show the results. From the result, we can see the differences in MMDs by change seed size are not significant in general. For some cases where MMD score increased, we observed that small graphs were generated, which are likely the main reason. We will include the discussion on revision.
> >
> > Table 1. Results on grid dataset with seed size 5.
> > |                   | deg.     | clus.     | orbit.    | GK      |
> > | -------------  | --------- | ---------- | --------- | -------- |
> > | GRAM       | 0.169   | 0.001    | 0.024  | 0.292  |
> > | GRAM-A    | 0.521   | 0.003   | 0.152   | 0.324  |
> > | GRAM-B    | 0.050   | 6e-4     | 0.019   | 0.287  |
> > | GRAM-AB  | 0.607   | 0.007   | 0.222   | 0.316  |
> >
> > Table 2. Results on lobster dataset with seed size 5.
> > |                    | deg.    | clus.     | orbit.    | GK      |
> > | --------------  | -------- | ---------- | --------- | --------- |
> > | GRAM        | 0.002  | 0.001   | 0.001   | 0.061   |
> > | GRAM-A    | 0.003  | 4e-5     | 4e-5     | 0.062   |
> > | GRAM-B    | 8e-4    | 5e-4     | 4e-4     | 0.053   |
> > | GRAM-AB  | 0.002  | 1e-5     | 2e-4     | 0.053   |
> >
> > Table 3. Results on community dataset with seed size 5.
> > |                   | deg.     | clus.     | orbit.    | GK     |
> > | -------------  | --------- | ---------- | --------- | -------- |
> > | GRAM       | 0.016  | 0.024    | 0.019   | 0.016  |
> > | GRAM-A   | 0.030   | 0.004   | 0.025   | 0.015  |
> > | GRAM-B   | 0.145   | 0.150   | 0.040   | 0.026  |
> > | GRAM-AB | 0.139   | 0.111   | 0.048   | 0.025  |
> >
> > Table 4. Results on B-A dataset with seed size 5.
> > |                    | deg.     | clus.     | orbit.    | GK     |
> > | --------------  | --------- | ---------- | --------- | -------- |
> > | GRAM        | 0.057   | 0.006    | 0.038   | 0.015 |
> > | GRAM-A    | 0.053   | 0.075    | 0.045   | 0.016  |
> > | GRAM-B    | 0.049   | 0.022    | 0.037   | 0.015  |
> > | GRAM-AB  | 0.056   | 0.115    | 0.056   | 0.018  |

---

### Official Review · AnonReviewer3 · 2019-10-25
**Official Blind Review #3**

**Rating:** 3

**Review:**

This paper presents new graph generative model (GRAM) which claims to tackle the scalability issue found is most of the published models. The propose architecture can scale on graphs from three perspective - number of graphs, number of nodes/edges and number of node/edge labels. This is achieved by sequentially generating the subgraph using graph attention and graph convolutional layers. While training, each of these subgraph can be trained in parallel. Further, paper applies couple of heuristics to reduce computational time and introduces a new non-domain specific evaluation metric for the generation of node/edge labeled graphs.

Although the paper claims to propose simplified mechanism, I find the generation task to be relatively very complex in comparison to GraphRNN and GRAN (published at NeurIPS'19). As mentioned below, the use of certain module seems ad-hoc. Further, the results on the new metric is at times inconsistent with other prior metrics. In lieu of this, currently the paper leans towards rejection. I would be happy to improve my score if some of the below-mentioned concerns are addressed.

Clarification:
1. What is the unit for time in Table 1 ? Is it inference time or training time ?
2. In my experience, the change in quantitative number do not necessarily reflect improvement in qualitative output. The metric GK is inconsistent. For example, on grid - GraphRNN is better on three metrics while GRAM gives the best results on GK. On community, there is a wide discrepancies between GraphRNN and GraphRNN-S model for most metrics but for GK.
3. Can you guide me on qualitative results of community and B-A graphs data ? Currently, nothing could be interpreted from these plots. Moreover, why the training set of community graph fails to show 4 commnities ? May be you should modify the data generation process.

Concern and Additional Experiments:
1. Please use standard Grid graph dataset as used in the literature - max |V| = 361. Moreover, I was wondering how do one generate 500 grid graphs with just max 100 nodes ? Since these graphs are not random.
2. Node scalability - GRAM has been employed only on graphs of maximum size 500 nodes. This does not confirm scalability. The advantage of parallel training of GRAM as against sequential GraphRNN should be showcased on large graphs of atleast 5000 nodes.
3. Please include results on newer models pub lished at NeurIPS'2019 - Graph Recurrent Attention Network (GRAN) and Graph Normalizing flows (GNF).
4. No one model among GRAM is projecting out to be best. On couple of data, GRAM is best, while on others GRAM-A or GRAM-B is better.
5. During inference, GRAM needs to compute the shortest path length among different nodes. This will surely not scale up with increasing nodes. Moreover, from Table 6 it is inconclusive whether that bias term is useful. How does the results look if both the biases are removed ?
6. I note that each input node vector stacks degree and clustering coefficient information. How one obtains this information during inference ? Yet again, it will face scalability issue as above.
7. The above concern also highlight the fact that the statistics of measured metrics (degree and clustering coefficients) are utilized during training. It seems more like a hack to me. No wonder this leads to performance boost of GRAM. Please share ablation study on this.
8. Please explain how Graph convolutional complements the processing in graph attention. Is both required ? Can you share ablation study on this ?

Minor:
1. The models categorized as unsupervised indeed trains using supervision of edge connectivity.
2. Frist -> First

**Experience Assessment:**

I have read many papers in this area.

**Review Assessment: Checking Correctness Of Derivations And Theory:**

I carefully checked the derivations and theory.

**Review Assessment: Checking Correctness Of Experiments:**

I carefully checked the experiments.

**Review Assessment: Thoroughness In Paper Reading:**

I read the paper thoroughly.

---

> ### Author Response · Authors · 2019-11-11
> **Author Response for Official Blind Review #3 (1/2)**
>
> To Reviewer #3
>
> Thank you for the detailed review and helpful suggestions.
> Below we answer the clarification points in the review one by one.
> We are preparing and conducting additional experiments/analysis now, and we will report the results as soon as ready.
>
> # “I find the generation task to be relatively very complex in comparison to GraphRNN and GRAN (published at NeurIPS'19).”
> Regarding the complexity of the generation task, this is because we target labeled graph generation tasks whereas GraphRNN and GRAN target unlabeled graph generation tasks. Considering node/edge labels requires additional operations such as node label estimation, edge label estimation, and feature extraction.
> Regarding the complexity of the proposed model, we are aware that our whole pipeline is relatively complex compared with GraphRNN and GRAN. However, one of the main differences between our model and them is that GraphRNN and GRAN do not consider node/edge labels in feature extraction and estimation. To consider node/edge labels, our architecture had to be complex accordingly.  Comparing DeepGMG, which considers node/edge labels,  we think the overall framework of our model is simpler enough.
> Additionally, it would like to be noted that GraphRNN-S and GRAN pose some strong assumptions such as independence between edges to simplify the generation process. In contrast, our main model is built on the likelihood formulation of graphs in Section 3.2 without any such strong assumptions, which is for not limiting model capacity.
> In addition, Our motivation to use the low-level graph statistics was for efficient feature extraction.  More precisely, we expected that node features output by the first feature extraction block will be low-level statistics like degree or clustering coefficient. Thus, we took a strategy to feed them by calculating them, which is more efficient than graph convolution and graph attention. We will specify that on revising.
>
> # Clarification 1
> The reported time in Table 1 is training time and the unit is hour. Although we wrote that at the below on page 7, we will specify them in Table 1 for clarity on revising.
>
> # Clarification 2
> The inconsistency between the topological MMD scores (degree, clustering coefficient, and orbit count) and GK-MMD scores is likely because GK-MMD considers node/edge labels and the formers do not. More precisely, GraphRNN is good at unlabeled graph generation tasks, while poor at considering node/edge labels. Thus, its topological MMD scores are good whereas the GK-MMD score is poor. In our opinion, node/edge labels also have to be considered in the quantitative evaluation.
>
> # Concern and Additional Experiments 4
> Yes, that is true. First, we would like to mention that in general GRAM variants demonstrated better than baselines on average. In addition, we would like the merit to be focused that these approximation variants can be selected according to the target graph properties, as discussed in Section A.5 and Section A.6. In these sections, we included discussions on possible reasons that approximation (GRAM-A and GRAM-B) not only reduces the computational cost but also boosts the performance.
>
> # Concern and Additional Experiments 5
> Yes, the computational cost for the shortest path length calculation depends on the graph size. As the generation process is sequential, it is possible to reuse the result in $n-1$-th step to calculate the shortest paths in $n$-th step, though currently, we do not use this for implementation. More precisely, we can keep all shortest path lengths of the previous step and update it on the next step by conducting a breadth-first search from the new node and comparing the path lengths. This will alleviate the scalability problem on the shortest path length calculation during the inference phase.
>
> # Concern and Additional Experiments 6
> We calculate these statistics on subgraphs generated at the step. Yes, its computational cost depends on the subgraph size. As for the shortest path length calculation, we believe that it is also possible to alleviate the problem by reusing the result on the previous subgraph.
>
> # Concern and Additional Experiments 7
> Our motivation to use the low-level graph statistics was for efficient feature extraction.  More precisely, we expected that node features output by the first feature extraction block will be low-level statistics like degree or clustering coefficient. Thus, we took the strategy to feed them by calculation degree and clustering coefficient, which is more efficient than graph convolution and graph attention. We will specify that on revising.
>
> # Concern and Additional Experiments 8
> We used graph convolution to consider edge labels connected to the target nodes. Specifically, since the graph attention only considers nodes, we used graph convolution to extract local connectivity information like edge labels. We will add the explanation on revising

---

> > ### Author Response · Authors · 2019-11-11
> > **Author Response for Official Blind Review #3 (2/2)**
> >
> > # Minor 1
> > We agree with the standpoint. However, considering the model does not use any additional labels on graphs and previous works (as GraphRNN) are categorized to unsupervised learning, we categorized it to unsupervised. We think our model is categorized as unsupervised autoregressive models such as PixelCNN [1].
> >
> > # Minor 2
> > Thank you for pointing out. We will fix it on revising.
> >
> > # Reference
> > [1] A. van den Oord, N. Kalchbrenner, and K. Kavukcuoglu. Pixel recurrent neural networks. In ICML, 2016.

---

> > > ### Author Response · Authors · 2019-11-15
> > > **Author Response 2 for Official Blind Review #3**
> > >
> > > # Concern and Additional Experiments 2
> > > We conducted an experiment on lobster graphs with 1000<=|V|<5000. The sizes of training, validation, and test set are all 100. Due to the large graph size and limited time, we trained only GRAM-B and limited the feature extraction to degree and clustering coefficient calculation. We evaluated the performance by MMD scores of degree, clustering coefficient, and orbit count.
> > >  The table below shows the result. Although it is not a strict comparison, we can see that the MMD scores are competitive to those when the maximum graph size is 100 in Section A.7.3.
> > >
> > > Table 1. Results on lobster dataset with 1k<=|V|<5k.
> > > |                 | deg.     | clus.     | orbit.    |
> > > | -----------   | --------- | ---------- | --------- |
> > > | GRAM-B  | 0.001   | 1e-6     | 1e-4     |
> > >
> > >
> > > # Concern and Additional Experiments 3
> > > GRAN targets unlabeled graph generation tasks while we target labeled graph generation tasks, where nodes and edges are assigned labels. In addition, the results of GRAN could not be included because the paper had not be published at the submission date.
> > > As for GNF, it also generates only unlabeled graphs. Also, the architecture is expected not to scale w.r.t. graph size because it belongs to a one-shot tensor generation class. Actually, the graphs used in the experiment have at most 20 nodes.  Instead, we will include them in Related Work on revision.
> > >
> > > # Concern and Additional Experiments 7
> > > We aimed at efficient feature extraction by using these low-level graph statistics. To examine the effect, we conducted an experiment removing these statistics.
> > > The tables below show the results.  Removing these statistics slightly degraded the performance in the grid and lobster dataset.
> > >
> > > Table 2. Results without degree and clustering coefficient information on grid dataset.
> > > |                      | deg.     | clus.     | orbit.    | GK      |
> > > | -----------------| ----------| ---------- | --------- | --------- |
> > > | GRAM          | 0.184   | 0.014   | 0.067   | 0.292   |
> > > | GRAM-A      | 0.499   | 9e-4     | 0.129    | 0.326  |
> > > | GRAM-B      | 0.066   | 8e-5     | 0.026    | 0.289  |
> > > | GRAM-AB    | 0.394   | 3e-4     | 0.153   | 0.295   |
> > >
> > > Table 3. Results without degree and clustering coefficient information on lobster dataset.
> > >
> > > |                    | deg.     | clus.     | orbit.    | GK       |
> > > | -------------   | --------- | ---------- | --------- | --------- |
> > > | GRAM        | 0.005   | 2e-4     | 7e-4     | 0.072   |
> > > | GRAM-A    | 0.002   | 3e-5     | 1e-4      | 0.054   |
> > > | GRAM-B    | 0.002   | 0.002    | 6e-5     | 0.062   |
> > > | GRAM-AB  | 0.001   | 5e-5     | 2e-4     | 0.056    |
> > >
> > > Table 4. Results without degree and clustering coefficient information on community dataset.
> > > |                   | deg.     | clus.     | orbit.    | GK      |
> > > | ------------   | --------- | ---------- | --------- | --------  |
> > > | GRAM       | 0.039   | 0.017    | 0.020   | 0.016  |
> > > | GRAM-A   | 0.017   | 0.011    | 0.021   | 0.015  |
> > > | GRAM-B   | 0.128   | 0.101   | 0.033   | 0.022   |
> > > | GRAM-AB | 0.087   | 0.073   | 0.027   | 0.023   |
> > >
> > > Table 5. Results without degree and clustering coefficient information on B-A dataset.
> > > |                    | deg.     | clus.     | orbit.    | GK      |
> > > | -------------   | --------- | ---------- | --------- | --------  |
> > > | GRAM        | 0.055   | 0.010    | 0.025  | 0.016   |
> > > | GRAM-A    | 0.052   | 0.008    | 0.022   | 0.016  |
> > > | GRAM-B    | 0.100   | 0.023    | 0.049   | 0.016  |
> > > | GRAM-AB  | 0.095   | 0.027    | 0.079   | 0.020  |

---

### Public Comment · ~Sayan_Ranu2 · 2019-10-13
**What is the impact on the quality if no seeds are used?**

This is an interesting piece of work. I notice one key difference from the existing learning-based graph generative models such as GraphRNN and DeepGMG. The authors state on page 17 that they do not start from a 0-node graph. Rather they need an input seed, which ranges from 5 to 12 nodes in their experiments. This assumption is somewhat ad hoc.

1. I would be interested in looking at the results when GRAM starts from scratch.
2. I also wonder if GRAM can start from scratch at all since it relies on graph convolutions to compute features. When generation starts from scratch, the neighborhood is empty and hence the computed features are likely to be of poor quality as the learned weights from training are based on complete neighborhood information whereas during generation you only have partial/no information. How does this impact the future predictions?
3. How important is the structure of the seed itself? Which seed would you prefer between a benzene ring (6 nodes) vs a path of 6 nodes in a molecule? How were the seeds chosen in the experiments? randomly?
4. In addition to the size of the seeds in terms of nodes, the authors should also state the size in terms of the number of edges.

---

> ### Author Response · Authors · 2019-11-11
> **Author Response for Public Review**
>
> Thank you for the detailed review.
> Below we answer the clarification points in the review one by one.
> We are preparing and conducting additional experiments/analysis now, and we will report the results as soon as ready.
>
> # 1, 2
> Thank you for pointing it out.  As the public reviewer mentions,  in Section 3.2 we formulated the likelihood of graphs as generating from one node for simplicity, whereas we started the generation process from seed graphs with a few nodes in practice. We will make it clear in the main part of the paper to avoid confusion on revising.
> We used seed graphs for two reasons. One comes from compatibility with feature extraction. As the public reviewer pointing out, basically graph convolution cannot be defined on graphs with one node. It requires graphs to contain at least two nodes and one edge. Thus the proposed model cannot start generation from scratch. The other comes from learning stability. The smaller the number of nodes in seed graphs, the higher the frequency that the model faces specific patterns of graphs in training.  We expected it would unstabilize the training, so we set the seed graph size so that diverse enough as well as small enough for evaluation.
> Although this would not be so important and relevant, we believe that generating from seed graphs has pros and cons compared with from-scratch models such as DeepGMG and GraphRNN. The cons are what the public reviewer mentions. The pros are, for example, we can force any desired substructures to be contained in generated graphs. This would be useful for drug discovery. Although we currently do not implement it, it would be possible to use multi-parts substructures.
>
> # 3
> For all datasets, we sampled seed graphs randomly.
>
> # 4
> Thank you for the comment. We will add a description of that on revising.

---

> > ### Author Response · Authors · 2019-11-15
> > **Author Response 2 for Public Review**
> >
> > # 1, 2
> > Please refer to (5) in the Author Response 2 for Official Blind Review #2.

---

### Decision · Program_Chairs · 2019-12-19

**Decision:**

Reject

**Comment:**

The paper proposed an efficient way of generating graphs.  Although the paper claims to propose simplified mechanism, the reviewers find that the generation task to be relatively very complex, and the use of certain module seems ad-hoc. Furthermore, the results on the new metric is at times inconsistent with other prior metrics. The paper can be improved by addressing those concerns concerns.